# Complex formation of APP with GABA_B receptors links axonal trafficking to amyloidogenic processing

Margarita C. Dinamarca[1], Adi Raveh[1], Andy Schneider[2], Thorsten Fritzius[1], Simon Früh [1], Pascal D. Rem[1], Michal Stawarski[1], Txomin Lalanne[1], Rostislav Turecek[1,5], Myeongjeong Choo[1], Valérie Besseyrias[1], Wolfgang Bildl[2], Detlef Bentrop[2], Matthias Staufenbiel[3], Martin Gassmann [1], Bernd Fakler[2,4], Jochen Schwenk [2,4] & Bernhard Bettler[1]

GABA_B receptors (GBRs) are key regulators of synaptic release but little is known about trafficking mechanisms that control their presynaptic abundance. We now show that sequence-related epitopes in APP, AJAP-1 and PIANP bind with nanomolar affinities to the N-terminal sushi-domain of presynaptic GBRs. Of the three interacting proteins, selectively the genetic loss of APP impaired GBR-mediated presynaptic inhibition and axonal GBR expression. Proteomic and functional analyses revealed that APP associates with JIP and calsyntenin proteins that link the APP/GBR complex in cargo vesicles to the axonal trafficking motor. Complex formation with GBRs stabilizes APP at the cell surface and reduces proteolysis of APP to Aβ, a component of senile plaques in Alzheimer's disease patients. Thus, APP/GBR complex formation links presynaptic GBR trafficking to Aβ formation. Our findings support that dysfunctional axonal trafficking and reduced GBR expression in Alzheimer's disease increases Aβ formation.

[1] Department of Biomedicine, Institute of Physiology, University of Basel, Klingelbergstr. 50/70, 4056 Basel, Switzerland. [2] Faculty of Medicine, Institute of Physiology, University of Freiburg, Hermann-Herder-Str. 7, 79104 Freiburg, Germany. [3] Department of Cellular Neurology, Hertie Institute for Clinical Brain Research, University of Tübingen, Otfried-Müller-Strasse 27, 72076 Tübingen, Germany. [4] Signalling Research Centers BIOSS and CIBSS, University of Freiburg, Schänzlestr. 18, 79104 Freiburg, Germany. [5] Present address: Institute of Experimental Medicine, ASCR, Vı´denska´ 1083, 14220 Prague 4-Krc, Czech Republic. These authors contributed equally: Margarita C. Dinamarca, Adi Raveh, Andy Schneider. Correspondence and requests for materials should be addressed to J.S. (email: jochen.schwenk@physiologie.uni-freiburg.de) or to B.B. (email: bernhard.bettler@unibas.ch)

GABA$_B$ receptors (GBRs) are key regulators of synaptic transmission in the brain[1,2]. Presynaptic GBRs inhibit the release of a variety of neurotransmitters while postsynaptic GBRs generate inhibitory K$^+$ currents that hyperpolarize the membrane and inhibit neuronal activity. There is evidence for a downregulation of presynaptic GBRs in response to neuronal activity[3–5] and in disease, including Alzheimer's disease (AD)[6,7], Fragile-X syndrome[8], epilepsy[9], and Parkinson disease[10]. Despite their importance for proper brain functioning the trafficking molecules controlling presynaptic GBR abundance are still unknown. Heterodimeric GB1a/2 and GB1b/2 receptors accumulate at excitatory terminals and in the somatodendritic compartment, respectively[1,11–14]. The GBR subunit GB1a contains two N-terminal sushi domains (SDs) that, when deleted, impair axonal localization and surface stability of the receptor. GB1a knock-out ($GB1a^{-/-}$) mice therefore exhibit a lack of axonal GBRs and a deficit in GBR-mediated inhibition of glutamate release[1,12]. Native GB1a/2 receptors co-purify with kinesin-1 adapters of the c-Jun N-terminal kinase-interacting protein (JIP) and calsyntenin (CSTN) protein families[15], in agreement with kinesin-1 motors mediating long-range vesicular transport of GB1a/2 receptors in axons[16]. As the SDs of GB1a face the luminal side of cargo vesicles, an as-yet unidentified transmembrane protein must link the SDs to cytoplasmic kinesin-1 motors. Proteomic analysis revealed several transmembrane proteins that selectively co-purify with GB1a/2 receptors and potentially link SDs to kinesin-1 motors, including the β-amyloid precursor protein (APP), the adherence-junction associated protein 1 (AJAP-1) and the PILRα-associated neural protein (PIANP)[15]. APP is the source of β-amyloid (Aβ) peptides, a hallmark of Alzheimer's disease (AD)[17–19]. AJAP-1 interacts with the E-cadherin-catenin complex at adherens junctions, which mediate adhesion between pre-and postsynaptic membranes[20]. PIANP shares sequence-similarity with AJAP-1, undergoes polarized sorting in epithelial cells and attenuates E-cadherin cleavage by γ-secretase[21].

In the present study, we analyzed the interaction of APP, AJAP-1, and PIANP with GB1a and addressed the role of these proteins in presynaptic GBR transport and expression. Proteomic approaches indicate that APP, AJAP-1, and PIANP participate in distinct GBR complexes. NMR studies identify sequence-related epitopes in APP, AJAP-1, and PIANP that bind to the N-terminal SD (SD1) of GB1a, with a rank order of affinities AJAP-1 > PIANP >> APP. Selectively APP links GB1a/2 receptors to vesicular trafficking and, when deleted, induces a significant deficit in GBR-mediated inhibition of glutamate release. Intriguingly, APP/GB1a complex formation not only mediates presynaptic GB1a/2 receptor trafficking but also limits the availability of APP for endosomal processing to Aβ. The association of presynaptic GBR expression with APP processing can explain pathological features observed in AD and suggests APP/GB1a complex stabilization as a promising therapeutic strategy.

## Results

**APP, AJAP-1, and PIANP assemble into distinct GBR complexes.** To investigate the interdependence of protein constituents in GBR complexes we analyzed the composition of affinity-purified GBRs from $GB1a^{-/-}$, $APP^{-/-}$, $AJAP-1^{-/-}$, $PIANP^{-/-}$, and $APP/AJAP-1^{-/-}$ double knock-out brains using quantitative mass spectrometry[15] (Fig. 1a, Source Data). Lack of GB1a largely prevented assembly of APP, APLP-2, AJAP-1, PIANP, ITM2B/C, CSTN-3, and Synaptotagmin-11 (Syt-11) into GBR complexes, indicating that these proteins directly or indirectly interact with presynaptic GBRs (Fig. 1a). Analysis of protein levels in $APP^{-/-}$, $AJAP-1^{-/-}$, $PIANP^{-/-}$, and $APP/AJAP-1^{-/-}$ mice showed that

APP, AJAP-1, and PIANP form independent GBR complexes (Fig. 1a). GBRs of $APP^{-/-}$ brains lacked APP, APLP-2, ITM2B/C, CSTN-3, and Syt-11. In contrast, GBRs of $AJAP-1^{-/-}$ and $PIANP^{-/-}$ brains selectively lacked the deleted protein. Interestingly, deletion of APP or AJAP-1 increased the amount of PIANP in GBR complexes, likely because of the increased availability of SDs for binding (Fig. 1a). GBRs in $APP/AJAP-1^{-/-}$ mice exhibited roughly additive changes in protein constituents when compared to individual knock-out mice, corroborating that APP and AJAP-1 assemble into distinct GBR complexes (Fig. 1a). Whole brain membranes of $APP^{-/-}$ mice showed an increased abundance of PIANP after genetic ablation of APP (Fig. 1b), suggesting that lack of APP frees SDs that bind and stabilize PIANP. Together, these results indicate that APP, AJAP-1, and PIANP form separate complexes with GB1a. Only the APP/GB1 complex binds CSTN-3, a protein implicated in vesicular trafficking[22] and synapse formation[23] that provides a potential link to axonal kinesin-1 motors.

**APP interacts with CSTN kinesin-1 adapters.** We isolated native APP, AJAP-1 and PIANP complexes in a series of multi-epitope affinity-purifications[15] to address whether these proteins interact with trafficking factors (Fig. 1c, Source Data). Quantitative analysis of affinity purifications by mass spectrometry confirmed that all three proteins co-assemble with GB1, GB2 and KCTDs into multiprotein complexes. However, selectively APP associated with several additional constituents of the GBR proteome, including APLP-2, ITM2B/C, Syt-11, NSG1/2, and the kinesin-1 adapters CSTN-1/-2/-3. JIP-1, a linker between kinesin-1 and APP[22], and JIP-3, a protein indirectly associated with GB1a[15], did not co-purify with APP in significant amounts, possibly because of dynamic interaction(s) or sterical hindrance by the antibodies used for affinity purification. Together, these proteomic experiments confirm that selectively APP links GBRs to the trafficking machinery.

**APP, AJAP-1 and PIANP bind with nanomolar affinities to SD1.** Deletion mapping revealed that the extracellular acidic domain (AcD) of APP encompassing amino acids 191–294 interacts with the N-terminal SD1 of GB1a (Fig. 2a, b). Detailed structural analysis of the complex by two-dimensional $^1$H-$^{15}$N heteronuclear single quantum coherence (HSQC) spectra delineated amino acids 202–219 in the purified AcD as the SD1 binding-site (Fig. 2c). The intrinsically disordered AcD exhibits low signal dispersion of the backbone amide protons (8.0–8.6 ppm) in the presence or absence of recombinant SD1/2 protein (Fig. 2c). The extracellular domains of AJAP-1 and PIANP exhibit sequence similarity with APP residues 202–219 in a stretch of six amino acids featuring a conserved WG motif preceded by hydrophobic residues (Fig. 2c). These six amino acids represent a crucial element of the binding interface, as shown by $^1$H–$^{15}$N HSQC spectra of AJAP-1 and PIANP with and without SD1/2 (Supplementary Fig. 1a). Subsequent mutagenesis confirmed that deletion of APP residues 202–219 or replacement of AJAP-1 residues 181–186 with alanine abolishes binding to GB1a (Fig. 2d). Surface binding assays indicated a rank order of SD binding affinities AJAP-1 ($K_D$ 6.4 ± 2.4 nM, mean ± s.e.m.) > PIANP (29.1 ± 5.5 nM) >> APP (187.6 ± 27.9 nM) (Supplementary Fig. 1b). APLP-2 completely lacks sequence similarity with the SD-binding site of APP (Fig. 2c), consistent with APLP-2 interacting with GB1a via APP (Fig. 1a, c). GBR activity did not significantly influence the amount of APP, AJAP-1, PIANP, and other GB1-interacting proteins co-immunoprecipitating with GB1a (Supplementary Fig. 2a, b). Likewise, GBR activity did not modify the bioluminescence resonance energy transfer (BRET)

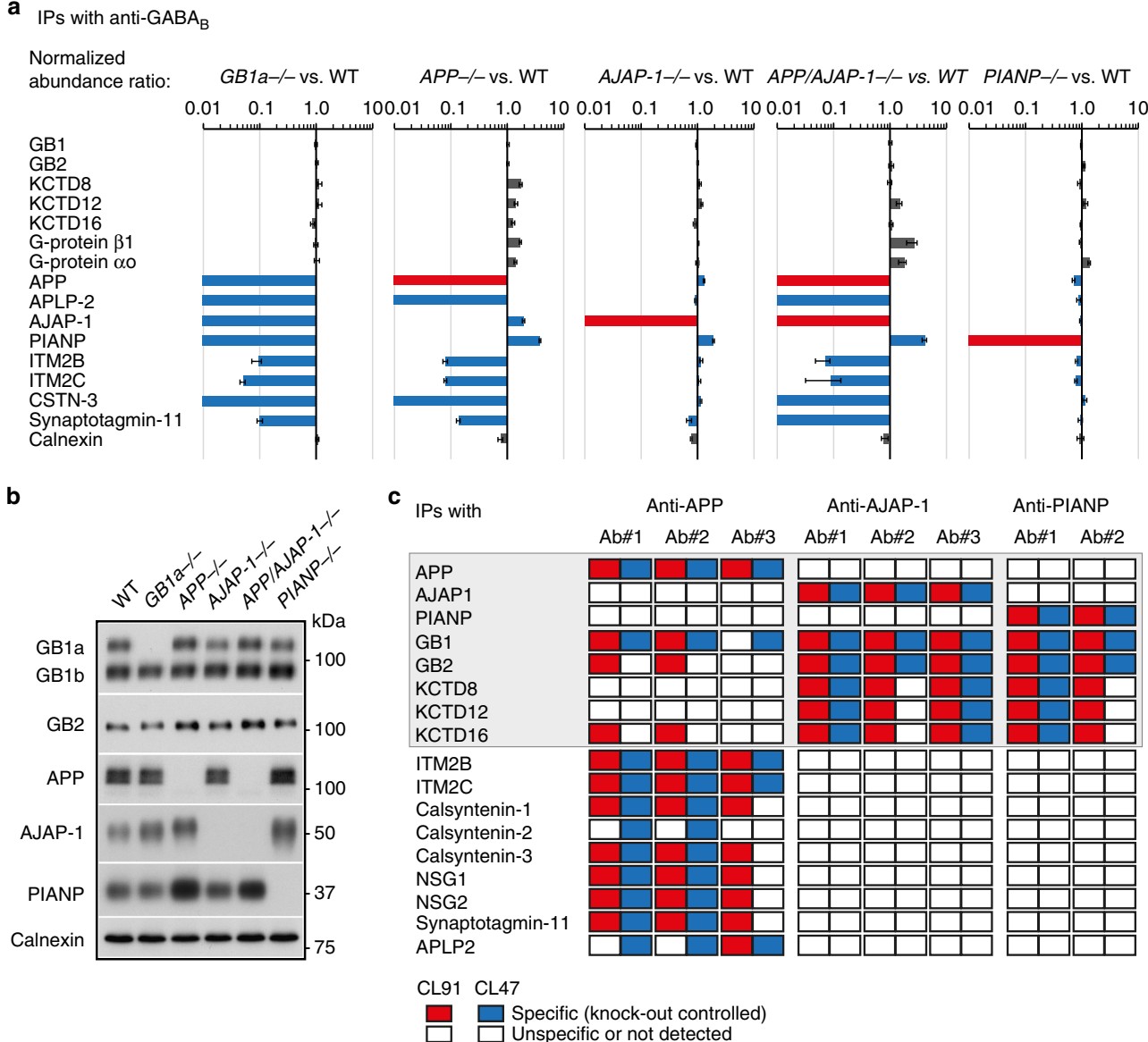

**Fig. 1** Proteomic analysis of native GBR complexes. **a** Protein abundance ratios for GBR proteome constituents ($n = 3$ measurements, data are presented as mean ± s.e.m.) in GBR IPs from membrane fractions of WT, $GB1a^{-/-}$, $APP^{-/-}$, $AJAP^{-/-}$, $APP/AJAP^{-/-}$, and $PIANP^{-/-}$ brains solubilized with the intermediate stringency detergent CL91 (normalized to GB1/2). APP, AJAP-1, and PIANP directly bind to GB1a while APLP2 binds to GB1a via APP. Common changes in the GBR proteomes of $GB1a^{-/-}$ and $APP^{-/-}$ mice identify APP as a putative linker between GB1a and the trafficking machinery. **b** Immunoblot analysis of GBR constituents in brain membrane preparations from knock-out mice. Calnexin serves as a loading control. Lack of APP upregulates PIANP. **c** APP, AJAP-1, and PIANP assemble into independent GBR complexes. The table summarizes results of co-immunoprecipitations with anti-APP, anti-AJAP-1, and anti-PIANP antibodies from mouse brain membranes solubilized with mild (CL47) and intermediate stringency (CL91) detergents. Knock-out brains were used in control immunoprecipitations. Source data are provided as a Source Data file

between APP-Venus and GB1a-Rluc in transfected HEK293 cells (Supplementary Fig. 2c, d).

**Lack of APP impairs presynaptic GBR-mediated inhibition.** We next addressed whether the lack of APP, AJAP-1, or PIANP impairs GBR-mediated inhibition of excitatory postsynaptic currents (EPSCs) at CA3/CA1 synapses. The prototypical GBR agonist baclofen was less efficient in reducing the amplitude of evoked EPSCs in $APP^{-/-}$ than in WT hippocampal slices (WT: 53.2 ± 2.7%, $APP^{-/-}$: 38.7 ± 2.8%; $P < 0.01$; Fig. 3a). There was also a trend towards reduced presynaptic inhibition of EPSC amplitudes in $AJAP-1^{-/-}$ and $PIANP^{-/-}$ slices, which however did not reach

significance ($P > 0.05$, Fig. 3a). Consistent with impaired baclofen-mediated inhibition of evoked EPSCs in $APP^{-/-}$ slices we also observed impaired baclofen-mediated inhibition of the miniature EPSC (mEPSC) frequency (WT: 64.4 ± 2.6%; $APP^{-/-}$: 43.9 ± 3.2%; $P < 0.001$; Fig. 3b). Baseline mEPSC frequency and amplitude were unaltered in $APP^{-/-}$ slices (Fig. 3b). Similarly, baclofen-mediated inhibition of the mEPSC frequency in cultured hippocampal neurons was impaired in $APP^{-/-}$ mice (WT: 74.07 ± 0.98%, $APP^{-/-}$: 57.01 ± 4.37%; $P < 0.001$), without a change of mEPSC amplitude ($P > 0.05$, Supplementary Fig. 3). Of note, the deficit in presynaptic inhibition in cultured $APP^{-/-}$ neurons was less pronounced than in $GB1a^{-/-}$ neurons (WT: 73.4 ± 1.7%,

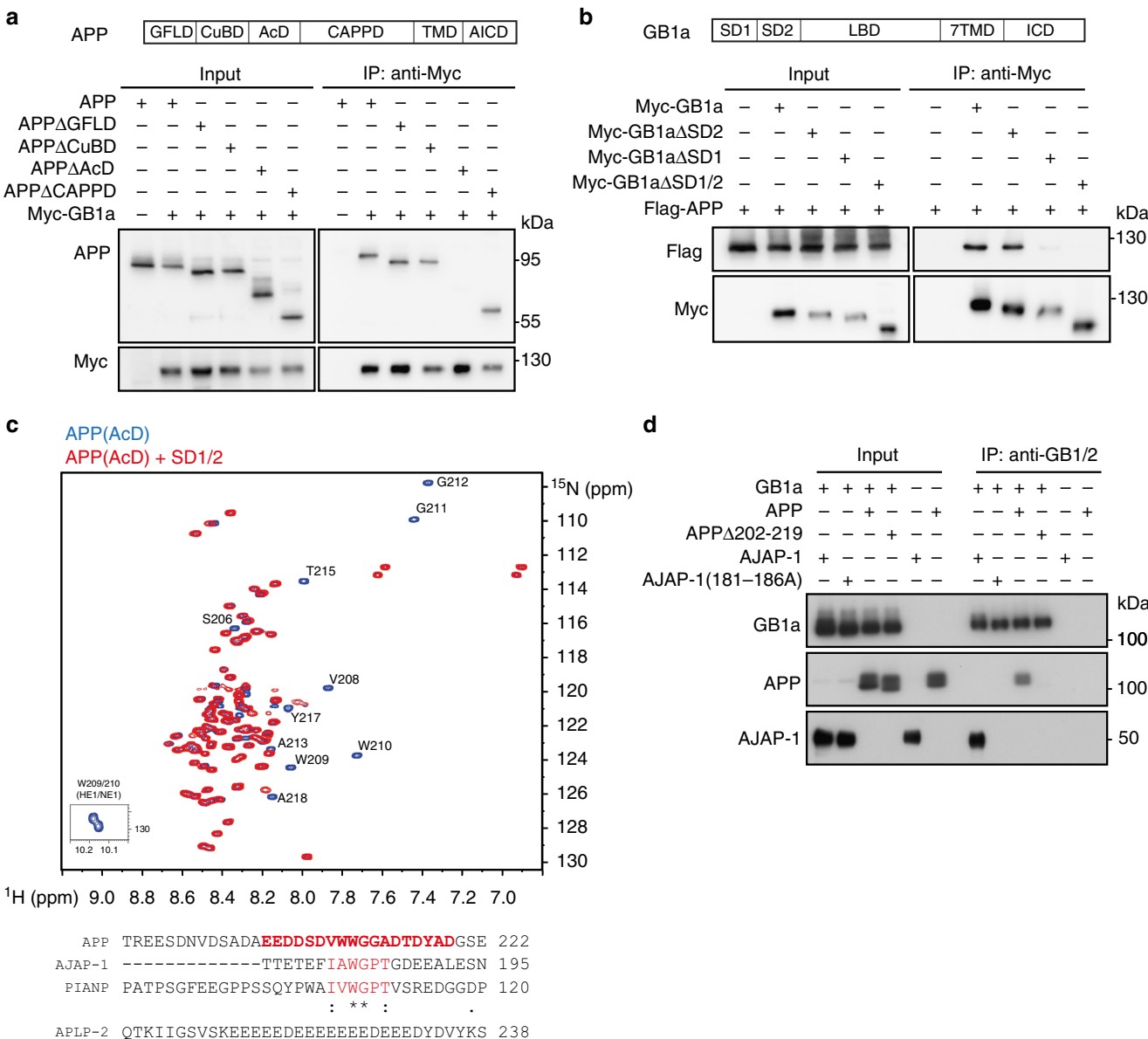

**Fig. 2** Interacting epitopes of APP, AJAP-1 and PIANP with GB1a. **a** The acidic domain (AcD) of APP comprising amino acids 191–294 is necessary for GB1a binding. Immunoprecipitations using anti-Myc-antibodies from HEK293 cells expressing Myc-GB1a together with APP or APP deletion mutants. Abbreviations: GFLD, growth factor-like domain; CuBD, copper-binding domain; CAPPD, central APP domain; TMD, transmembrane domain; AICD, APP intracellular domain. **b** The N-terminal SD1 is necessary for binding of APP. Immunoprecipitations from HEK293 cells expressing Flag-APP together with GB1a deletion mutants lacking SD1, SD2 or both SD1/2. **c** The AcD of APP interacts with recombinant SD1/2 protein via amino acid residues 202–219 containing a WG sequence motif. Top: Two-dimensional $^1$H–$^{15}$N heteronuclear single quantum coherence (HSQC) spectra of $^{13}$C/$^{15}$N labeled APP (AcD residues 191–294), alone (blue) or in complex (red) with unlabeled recombinant SD1/2. In complex with SD1/2 several APP residues exhibit chemical shift changes or disappear. These residues participate in protein-protein interaction (red in the sequence alignment). Amino acid assignment of APP was performed from a standard set of 3D experiments using $^{13}$C–$^{15}$N labeled AcD[65]. Bottom: Alignment of the APP epitope with related sequences in AJAP-1 and PIANP (red). APLP-2 exhibits no sequence similarity with the binding epitope of APP. **d** Deletion or mutation to alanine of the binding epitopes in APP and AJAP-1 prevents binding to GB1a, as shown in co-immunoprecipitation experiments. Source data are provided as a Source Data file

$GB1a^{-/-}$: 29.8 ± 2.5%; $P < 0.0001$; Supplementary Fig. 3), suggesting that a fraction of GBRs traffics to axon terminals in the absence of APP.

**APP does not influence GBR dependent G protein signaling.** It is conceivable that lack of APP impairs presynaptic GBR-mediated inhibition because APP normally exerts a positive allosteric effect on receptor-induced G protein signaling. We

carried out BRET experiments[24] in transfected HEK293 cells to analyze whether APP influences conformational changes of the G protein during GB1a/2 receptor activation. APP influenced neither the baseline BRET nor the magnitude or kinetics of the BRET change during G protein activation (Supplementary Fig. 4a). Because soluble extracellular APP fragments (sAPPα) are reported to signal through G proteins[17,25,26] we additionally incubated GB1a/2 receptors expressed in HEK293 cells with conditioned medium containing recombinant sAPPα

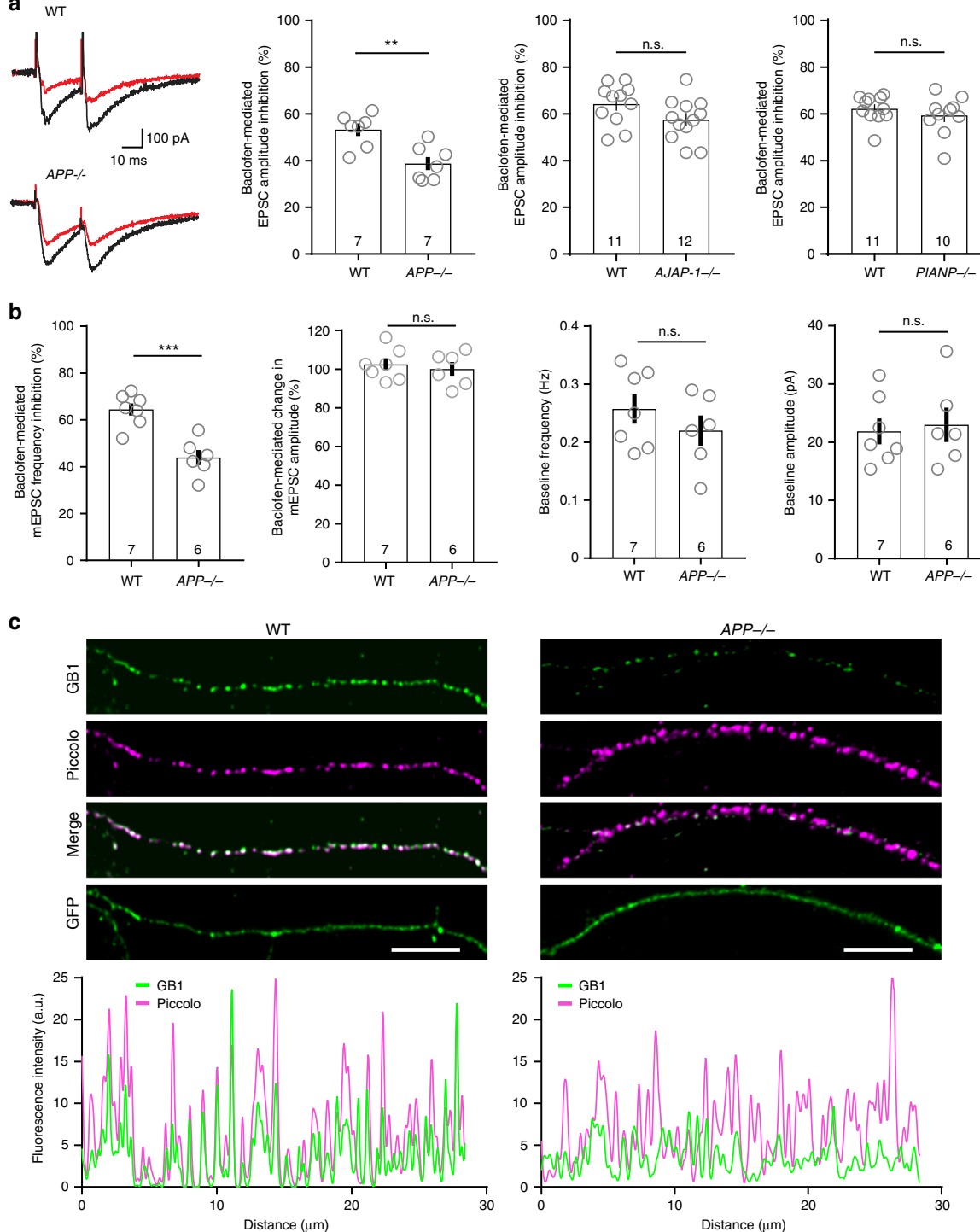

**Fig. 3** Reduced axonal GBR expression and presynaptic inhibition in $APP^{-/-}$ mice. **a** Representative traces of evoked EPSC recordings in CA1 neurons of acute hippocampal slices of $APP^{-/-}$ mice under baseline conditions (control, black) and during 50 μM baclofen application (red). Bar graphs show reduced baclofen-mediated EPSC amplitude inhibition in $APP^{-/-}$ mice (**$P < 0.01$, unpaired Student's $t$-test) while EPSC amplitude inhibition in $AJAP$-$1^{-/-}$, $PIANP^{-/-}$ mice and WT littermate mice did not differ ($P > 0.05$). The $n$ number of neurons is indicated. **b** Bar graphs showing reduced baclofen-mediated inhibition of the mEPSC frequency in CA1 pyramidal neurons of $APP^{-/-}$ mice (***$P < 0.001$, unpaired Student's $t$-test). Baclofen had no effect on the mEPSC amplitude in $APP^{-/-}$ and WT littermate mice. Baseline mEPSC frequency and amplitude were similar in $APP^{-/-}$ and WT littermate mice ($P > 0.05$, unpaired $t$-test). **c** Top: Immunofluorescence of endogenous GB1 protein in axons of hippocampal WT (left) and $APP^{-/-}$ (right) neurons. Neurons expressing GFP were fixed at DIV10, permeabilized, and immunostained for endogenous GB1 (green) and the presynaptic marker piccolo (magenta). GFP served as a volume marker. Merged images show GB1 and piccolo co-localization. Scale bar 5 μm. Bottom: Intensity gray value profile graphs of GB1 (green) and piccolo (magenta). Source data are provided as a Source Data file. Data are presented as mean ± s.e.m

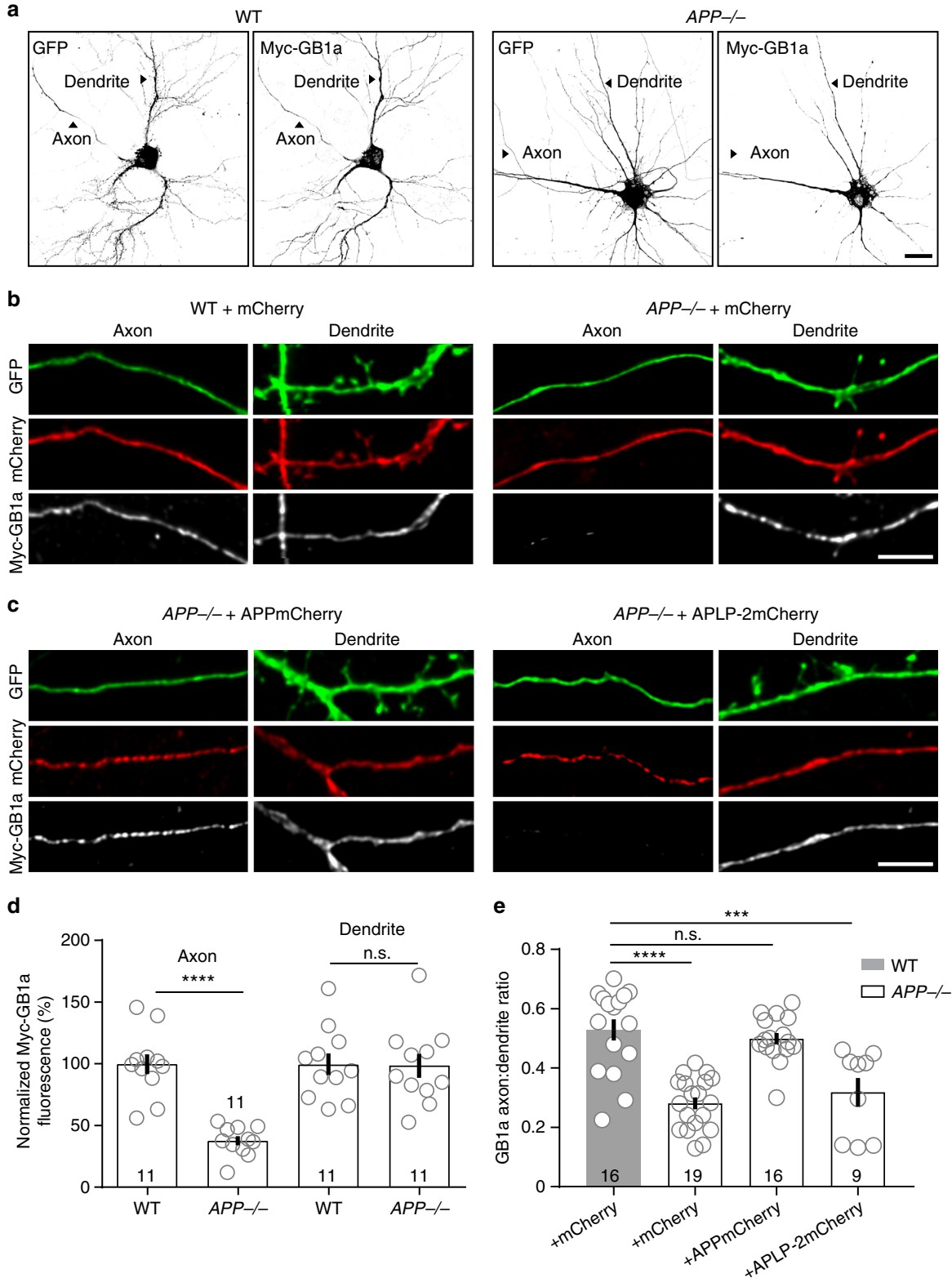

(Supplementary Fig. 4b). BRET analysis revealed no G protein activation upon sAPPα application. Brain membranes of $APP^{-/-}$ and WT littermate mice exhibited no difference in GABA-stimulated GTPγ[$^{35}$S] binding, corroborating that APP does not influence receptor-induced G protein signaling (Supplementary Fig. 4c). BRET and GTPγ[$^{35}$S] binding experiments therefore support that APP and sAPP do not modulate GBR activity.

**Reduced endogenous GB1 protein in axons of $APP^{-/-}$ neurons.** Lack of APP influence on GBR-induced G protein signaling suggests that reduced receptor numbers underlie impaired pre-synaptic inhibition in $APP^{-/-}$ neurons. We therefore determined endogenous GB1 expression in the axons of cultured hippo-campal $APP^{-/-}$ neurons using immunofluorescence staining (Fig. 3c). Normalization of GB1 immunofluorescence to that of

**Fig. 4** APP localizes exogenous GB1a protein to axons in cultured hippocampal neurons. **a** Representative images of hippocampal neurons co-expressing Myc-GB1a and GFP in $APP^{-/-}$ and control littermate (WT) mice. Neurons were transfected at DIV5, fixed at DIV10, permeabilized and then stained with anti-Myc antibodies. Note that $APP^{-/-}$ neurons exhibit significantly reduced axonal Myc-GB1a expression. Dendrites were distinguished from axons using morphological criteria[66]. Scale bar 10 μm. **b** Higher magnification images of distal axons and dendrites from $APP^{-/-}$ and WT neurons expressing exogenous Myc-GB1a, GFP and mCherry. Scale bar 10 μm. **c** Images of distal axons and dendrites from $APP^{-/-}$ neurons expressing exogenous Myc-GB1a, GFP and APPmCherry or APLP-2mCherry. Note that APPmCherry but not APLP-2mCherry rescues axonal localization of Myc-GB1a. Scale bar 10 μm. **d** Exogenous Myc-GB1a levels in axons and dendrites of transfected $APP^{-/-}$ or WT neurons. Normalized fluorescence refers to the Myc-GB1a immunofluorescence intensity normalized to the GFP fluorescence intensity. ****$P < 0.0001$, unpaired Student's $t$-test. **e** Axon:dendrite (A:D) ratio of Myc-GB1a in $APP^{-/-}$ and WT neurons transfected with Myc-GB1a in the presence of mCherry, APPmCherry or APLP-2mCherry (DIV10). The $n$ number of neurons analyzed is indicated. ***$P < 0.001$, ****$P < 0.0001$, one-way ANOVA. Data are presented as mean ± s.e.m. Source data are provided as a Source Data file

the volume marker GFP revealed a 74% reduction in GB1 expression in $APP^{-/-}$ axons (WT: 100 ± 10.1%, $APP^{-/-}$: 26.0 ± 3.7%; $n = 10$ neurons, 6 independent transfections per group, $P < 0.0001$, unpaired $t$-test). In WT and $APP^{-/-}$ axons 48 and 25%, respectively, of the GB1 clusters co-localized with piccolo, a protein associated with the presynaptic cytoskeleton (Fig. 3c). As a control, normalization of piccolo staining to GFP fluorescence revealed no significant difference between genotypes (WT: 22.7 ± 3.5 a.u.; $APP^{-/-}$: 16.6 ± 2.1 a.u.; $n = 10$ neurons, 6 independent transfections per group, $P > 0.05$, unpaired $t$-test). $APP^{-/-}$ neurons therefore exhibit reduced GB1 expression in axons and putative presynaptic structures. However, GB1 protein is still detectable in $APP^{-/-}$ axons, in agreement with electrophysiological data indicating residual GBR-mediated presynaptic inhibition in $APP^{-/-}$ neurons (Fig. 3a, b). $AJAP-1^{-/-}$ and $PIANP^{-/-}$ mice exhibit normal levels of endogenous GB1 protein in the axons of cultured hippocampal neurons (Supplementary Fig. 5), explaining why presynaptic GBR-mediated inhibition in these mice is normal (Fig. 3a).

**Rescue of axonal GBR localization in $APP^{-/-}$ neurons.** We next studied the localization of N-terminally tagged Myc-GB1a in cultured hippocampal $APP^{-/-}$ neurons. We transfected Myc-GB1a or control Myc-GB1b together with the volume-markers GFP and mCherry and determined the subunit distribution by immunostaining[13]. In WT neurons Myc-GB1a was present in axons, somata and dendrites (Fig. 4a, b) while Myc-GB1b was mostly excluded from axons (Supplementary Fig. 6a, b), as reported earlier[13]. To quantify axonal and dendritic Myc-GB1a and Myc-GB1b expression, we normalized the anti-Myc fluorescence to GFP fluorescence. Axonal Myc-GB1a expression was reduced by 64% in $APP^{-/-}$ neurons (WT: 100 ± 7.3%, $APP^{-/-}$: 36.4 ± 3.6%, $n = 12$ neurons; $P < 0.0001$) while dendritic Myc-GB1a expression remained normal (WT: 100 ± 8.8%, $APP^{-/-}$: 98.7 ± 9.7%: $n = 11$ neurons, $P > 0.05$; unpaired Student's $t$-test; Fig. 4d). Accordingly, the axon-to-dendrite (A:D) ratio of Myc-GB1a was significantly reduced in $APP^{-/-}$ neurons (WT: 0.53 ± 0.04, $APP^{-/-}$ 0.28 ± 0.02; $P < 0.0001$; Fig. 4e). In contrast, axonal Myc-GB1b expression and the A:D ratio (WT: 0.34 ± 0.03, $APP^{-/-}$ 0.30 ± 0.03; $P > 0.05$) were similar in $APP^{-/-}$ and WT neurons and markedly lower than for Myc-GB1a (Supplementary Fig. 6c, d). We examined whether exogenous APP expression in $APP^{-/-}$ neurons rescues axonal localization of Myc-GB1a. We co-transfected cultured hippocampal neurons from $APP^{-/-}$ mice with mCherry-tagged APP (APPmCherry) and Myc-GB1a or Myc-GB1b, together with GFP as a volume marker (Fig. 4c, Supplementary Fig. 6d). APPmCherry significantly increased the A:D ratio of Myc-GB1a compared to control transfections with mCherry alone (+mCherry: 0.28 ± 0.02, +APPmCherry: 0.50 ± 0.02; $P < 0.0001$; Fig. 4c, e). In contrast, transgenic expression of

APPmCherry in $APP^{-/-}$ neurons had no effect on the A:D ratio of Myc-GB1b (+mCherry: 0.30 ± 0.03, +APPmCherry: 0.31 ± 0.02; $P > 0.05$; Supplementary Fig. 6d). Similarly, APLP-2mCherry did not rescue axonal localization of Myc-GB1a (Fig. 4c) and had no significant effect on the A:D ratio of GB1a in $APP^{-/-}$ neurons (+mCherry: 0.28 ± 0.02, +APLP-2mCherry: 0.32 ± 0.05; $P > 0.05$; Fig. 4e). These experiments identify APP as a key factor mediating axonal localization of GBRs.

**Visualization of APP/GB1a complexes in axons and dendrites.** We used bimolecular fluorescence complementation[27] (BiFC) to investigate APP/GB1a complex localization in axons, after tagging the two proteins at their C-termini with the N-terminal and C-terminal fragments of fluorescent Venus protein (VN, VC; Fig. 5a). BiFC in transfected HEK293 cells was successful between APP-VN and GB1a-VC but not between APP-VN and GB1b-VC (Supplementary Fig. 7a). BiFC imaging in HEK293 cells using total internal reflection fluorescence (TIRF) microscopy showed that the APP-VN/GB1a-VC complex is assembled early in the biosynthetic pathway in the perinuclear region and requires GB2 for surface expression[1] (Supplementary Fig. 7b). Transfected cultured hippocampal neurons exhibited APP-VN/GB1a-VC BiFC in axons, soma and dendrites (Fig. 5a). Co-expression of APP-VN with GB1b-VC generated no BiFC (Fig. 5a) even though the fusion proteins expressed normally (Supplementary Fig. 7c). In axons, the APP-VN/GB1a-VC complex partly co-localized with piccolo (Fig. 5b, correlation coefficients for BiFC/piccolo: Pearson 0.69 ± 0.02, Mander's 0.55 ± 0.03, $n = 11$ neurons, 4 independent transfections per group). The APP-VN/GB1a-VC complex also co-localized with the co-expressed presynaptic marker Synaptophysin-mCherry at boutons opposing PSD-95-positive spines (Supplementary Fig. 8). In dendrites, the APP-VN/GB1a-VC complex was present in the shafts but excluded from the spines (Fig. 5b), as shown by lack of co-localization with PSD-95 (overlap coefficient BiFC with PSD-95: Pearson 0.05 ± 0.01, Mander's 0.04 ± 0.02, $n = 11$ neurons, 4 independent transfections per group). This agrees with earlier findings showing that GB1a is not entering dendritic spines[3,12].

**APP/GB1a complexes traffic in axons.** Our proteomic data suggest that members of the CSTN and JIP family of proteins, which are kinesin-1 adapters[22,28], associate with GB1/2 receptors through APP[15] (Fig. 1). We studied whether APP-VN/GB1a-VC complexes (identified by BiFC) co-localize with CSTN-1/-3 and JIP-1/-3 in the axons of transfected hippocampal neurons (Fig. 5c, Supplementary Fig. 9). Mander's overlap coefficient analysis indicated that a fraction of APP-VN/GB1a-VC complexes co-localizes with CSTN-1/-3 and JIP-1/-3 (Fig. 5d). Moreover, the APP-VN/GB1a-VC complex partly co-localized with endogenous kinesin light-chain 1 (KLC1) protein (Fig. 5c, Supplementary Fig. 9), consistent with reports on CSTN and JIP family members

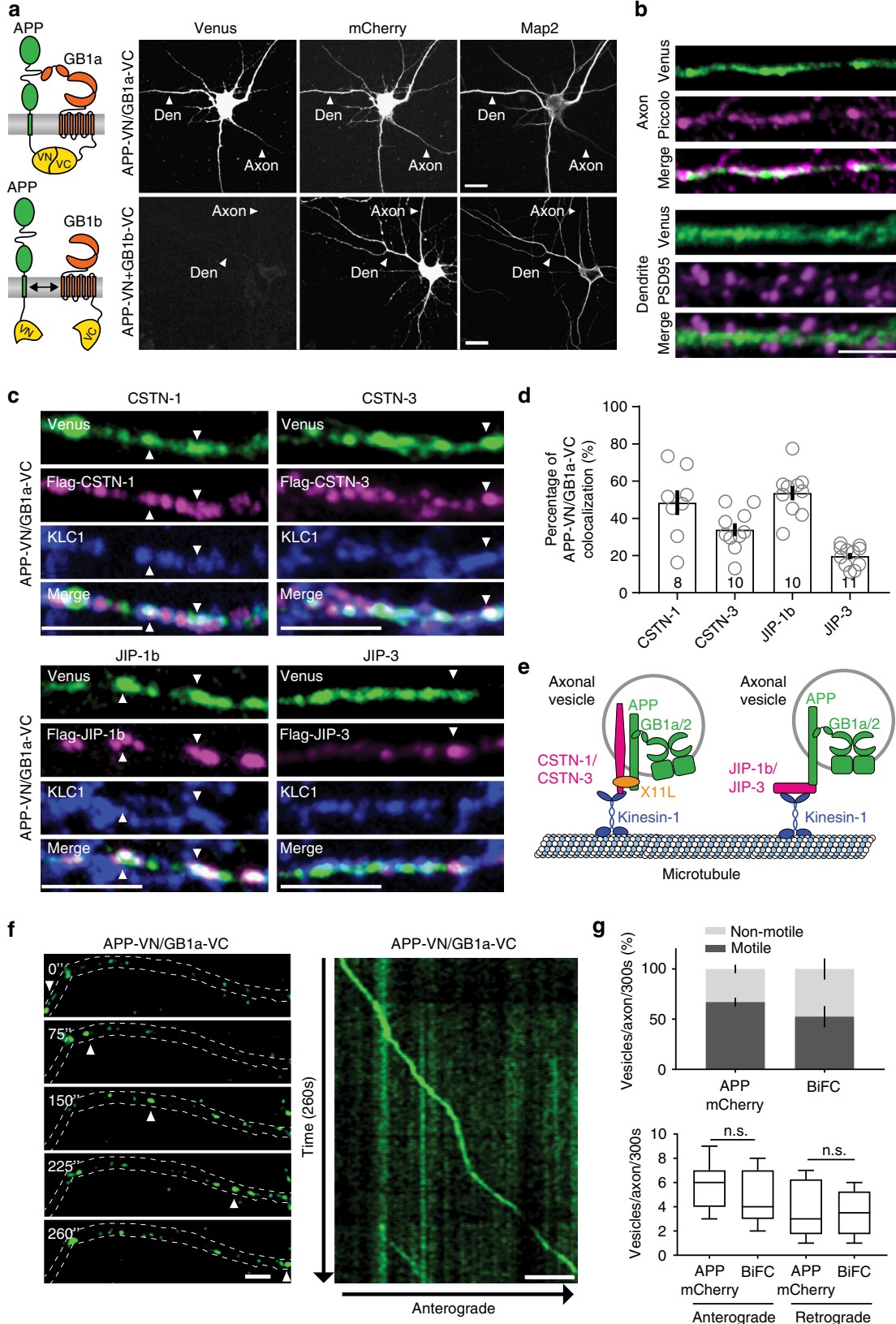

linking APP to kinesin motors[22,28] (Fig. 5e). We used live-cell confocal imaging to visualize axonal trafficking of the APP-VN/GB1a-VC complex and APPmCherry in cultured hippocampal neurons (Fig. 5f, Supplementary movie 1). Kymographs revealed

anterograde and retrograde axonal transport of APPmCherry and APP-VN/GB1a-VC BiFC complexes (Fig. 5g, Supplementary Fig. 10a), similarly as previously observed for APP[29]. The majority of APP vesicles colocalized with NPY, a marker for

**Fig. 5** The APP/GB1a complex co-localizes with CSTN and JIP proteins in axons. **a** Scheme depicting the BiFC principle[27]. Complex formation of APP-VN with GB1a-VC reconstitutes Venus fluorescence and leads to BiFC. GB1b-VC serves as a negative control. Representative confocal images show hippocampal neurons (DIV10) expressing APP-VN together with GB1a-VC or GB1b-VC. BiFC is observed in axons and dendrites for GB1a-VC. Microtubule-associated protein Map2 identifies dendrites; mCherry served as a volume marker. Neurons were imaged 7 h post-transfection[27]. Scale bar 10 µm. **b** Higher magnification of axons and dendrites of hippocampal neurons transfected with APP-VN and GB1a-VC. The BiFC complex (Venus) partly co-localizes with piccolo (magenta) in axons. The BiFC complex is also present along dendritic shafts but excluded from spines (PSD-95, magenta). Scale bar 5 µm. **c** Partial co-localization (white, arrowheads) of the BiFC complex (green) with FLAG-CSTN-1, FLAG-CSTN-3, FLAG-JIP-1b, and FLAG-JIP-3 (magenta) and endogenous kinesin light-chain 1 (KLC1) (blue) in the axons of neurons. Scale bar 5 µm. **d** Quantification of the co-localization of the BiFC complex with FLAG-CSTN-1, FLAG-CSTN-3, FLAG-JIP-1b, and FLAG-JIP-3. The n numbers of neurons analyzed are indicated. **e** Scheme illustrating that APP together with interacting JIP and CSTN proteins link the GB1a/APP complex in cargo vesicles to axonal kinesin-1 motors. The neural adaptor protein X11-like (X11L) connects APP to CSTN-1[22]. **f** Time-lapse images of a well-separated APP-VN/GB1a-VC complex trafficking anterogradely in axons (acquisition times in seconds). White arrowheads mark a fluorescent APP-VN/GB1a-VC complex. A kymograph shows the entire time-lapse recording (right). Scale bars 25 µm. Data are presented as mean ± s.e.m. **g** Top: Analysis showing the percentage of mobile and non-mobile vesicles per axon within 5 min in hippocampal neurons expressing APPmCherry or the BiFC complex. Bottom: Number of vesicles moving antero-gradely and retrogradely per axons within 5 min. Data are presented in a min to max-box and whisker plot, with whiskers representing the smallest and largest values, the boxes representing the 25–75% percentile and the middle line representing the median. $P > 0.05$, one-way ANOVA. Source data are provided as a Source Data file

Golgi-derived vesicles[29]. 51% GB1a-GFP and 54% APP-VN/GB1a-VC axonal vesicles also contained NPY-mCherry (Supplementary Fig. 11). The percentage of mobile APPmCherry vesicles in axons was similar to that reported for APP vesicles in cultured hippocampal/cortical neurons[30] and similar to that of APP-VN/GB1a-VC (Fig. 5g) or APPmCherry/GB1a-GFP vesicles (Supplementary Fig. 10b). However, significantly less GB1a-GFP (23%) than APPmCherry (73%) or APPmCherry/GB1a-GFP (66%) vesicles were mobile (Supplementary Fig. 10b), suggesting that endogenous APP is limiting for axonal transport of over-expressed GB1a-GFP. The fraction of vesicles trafficking antero-gradely or retrogradely in axons was similar for APP-VN/GB1a-VC and APPmCherry (Fig. 5g, Supplementary Fig. 10b). The mean anterograde ($2.2 \pm 0.4$ µm/s, $n = 29$ vesicles) and retrograde ($2.0 \pm 0.2$ µm/s, $n = 29$) trafficking velocities of APPmCherry in axons were similar as reported for dendrites[29]. APPmCherry/GB1a-GFP (antero: $1.6 \pm 0.2$ µm/s, $n = 16$: retro: $1.3 \pm 0.2$ µm/s, $n = 26$) and APP-VN/GB1a-VC vesicles (antero: $1.6 \pm 0.1$, $n = 19$; retro: $1.5 \pm 0.1$ µm/s, $n = 8$) had lower mean transport velocities than APPmCherry alone. GB1a-GFP vesicles (anterograde: $1.0 \pm 0.1$, $n = 15$; retrograde: $1.2 \pm 0.1$ µm/s, $n = 25$) had even lower transport velocities, similarly as observed before[16].

**Interaction of APP with GB1a inhibits Aβ generation.** Similar to other APP interacting proteins[22], GB1a/2 receptors may influence continuous proteolytic processing of APP. We investigated whether GB1a protects APP from cleavage by the β-site APP cleaving enzyme (BACE1) that, together with γ-secretase, generates Aβ[18] (Fig. 6a). Immunoblot analysis of transfected HEK293 cells indeed showed that co-expression of GB1a/2 receptors with APP markedly reduced the BACE1-cleavage products sAPPβ and β-carboxy-terminal fragment (APP-βCTF). Densitometric analysis revealed a reduction in the APP-βCTF/APP-FL and sAPPβ/APP-FL ratio by 60% ($P < 0.0001$ vs. exogenous APP together with BACE1) and 57% ($P < 0.01$), respectively, in the presence of GB1a/2 receptors (Fig. 6b). GB1b/2 receptors had no significant effect on BACE1 cleavage (Fig. 6b). GB1a/2 receptors had no effect on APP processing by ADAM10, an enzyme involved in non-amyloidogenic processing of APP (Supplementary Fig. 12a). We also studied amyloid-β40 (Aβ40) production in HEK293 cells expressing APP with GB1a/2 or GB1b/2, together with BACE1 or ADAM10 (Fig. 6c). The amount of Aβ40 secreted into the conditioned medium was determined 32 h post-transfection using a commercial ELISA. Expression of BACE1 but not ADAM10 increased Aβ40 secretion by one order

of magnitude. The presence of GB1a/2 receptors reduced this BACE1-mediated Aβ40 secretion by 77% ($P < 0.01$ vs. exogenous APP with BACE1). A smaller non-significant decrease in Aβ40 secretion was also observed in the presence of GB1b/2 receptors ($P > 0.05$ vs. exogenous APP with BACE1).

**GB1a/2 receptors stabilize APP at the cell surface.** GBRs activity did not influence Aβ40 secretion in cultured hippocampal neurons (Fig. 6d), consistent with GBR activity not influencing the APP/GB1a interaction (Supplementary Fig. 2). However, cultured hippocampal neurons of $GB1a^{-/-}$ mice exhibited a significant increase in secreted Aβ40 protein ($P < 0.0001$ vs. WT, Fig. 6e) indicating that loss of GB1a promotes amyloidogenic processing of APP. Control neurons of $GB1b^{-/-}$ mice exhibited no increase in Aβ40 secretion ($P > 0.05$ vs. WT, Fig. 6e). Infection of cultured hippocampal neurons of $GB1a^{-/-}$ or WT mice with lentiviral particles expressing GFP-GB1a significantly reduced Aβ40 secretion ($P < 0.05$ vs. GFP or GFP-GB1b, Fig. 6f). We investigated how GB1a/2 receptors inhibit amyloidogenesis. Immunoprecipitation experiments with transfected HEK293 cells showed that GB1a does not compete with BACE1 for APP (Supplementary Fig. 12b). Surface biotinylation experiments with cultured hippocampal neurons indicated that loss of GB1a but not GB1b significantly ($P < 0.01$) reduces APP levels (Fig. 7a), suggesting that GB1a stabilizes APP at the plasma membrane. Consistent with this hypothesis, surface biotinylation experiments in transfected HEK293 cells confirmed that GB1a/2 receptors significantly ($P < 0.05$) increased APP at the cell surface while GB1b/2 receptors failed to exert such an effect ($P > 0.05$, Fig. 7b).

We next investigated whether increased APP surface expression in the presence of GB1a reflects reduced endocytosis. We fused a 13 amino-acid α-bungarotoxin (BTX) binding site (BBS) to the N-terminus of APPmCherry[27] (Fig. 7c). TIRF microscopy was used to analyze time-dependent changes in surface fluorescence of BTX-488 (Alexa488-labeled BTX) and mCherry in HEK293 cells expressing BBS-APPmCherry in the absence or presence of GB1a/2 or GB1b/2 receptors (Fig. 7c). Endocytosis decreased BTX-488 cell surface fluorescence during a 15 min period, both in the absence of GBRs (residual TIRF intensity $58.9 \pm 3.1\%$) and in the presence of GB1b/2 receptors ($59.5 \pm 5.5\%$). GB1a/2 receptors significantly reduced the decrease in BTX-488 cell surface fluorescence ($81.0 \pm 2.8\%$, $P < 0.001$), indicating that association with GB1a/2 receptors reduces APP endocytosis (Fig. 7c). Of note, the cell surface mCherry fluorescence remained unaltered during the internalization period (control $103.7 \pm 7.0\%$, GB1a/2 $101.0 \pm 5.1\%$, GB1b/2 $112.6 \pm 8.0\%$, normalized to the

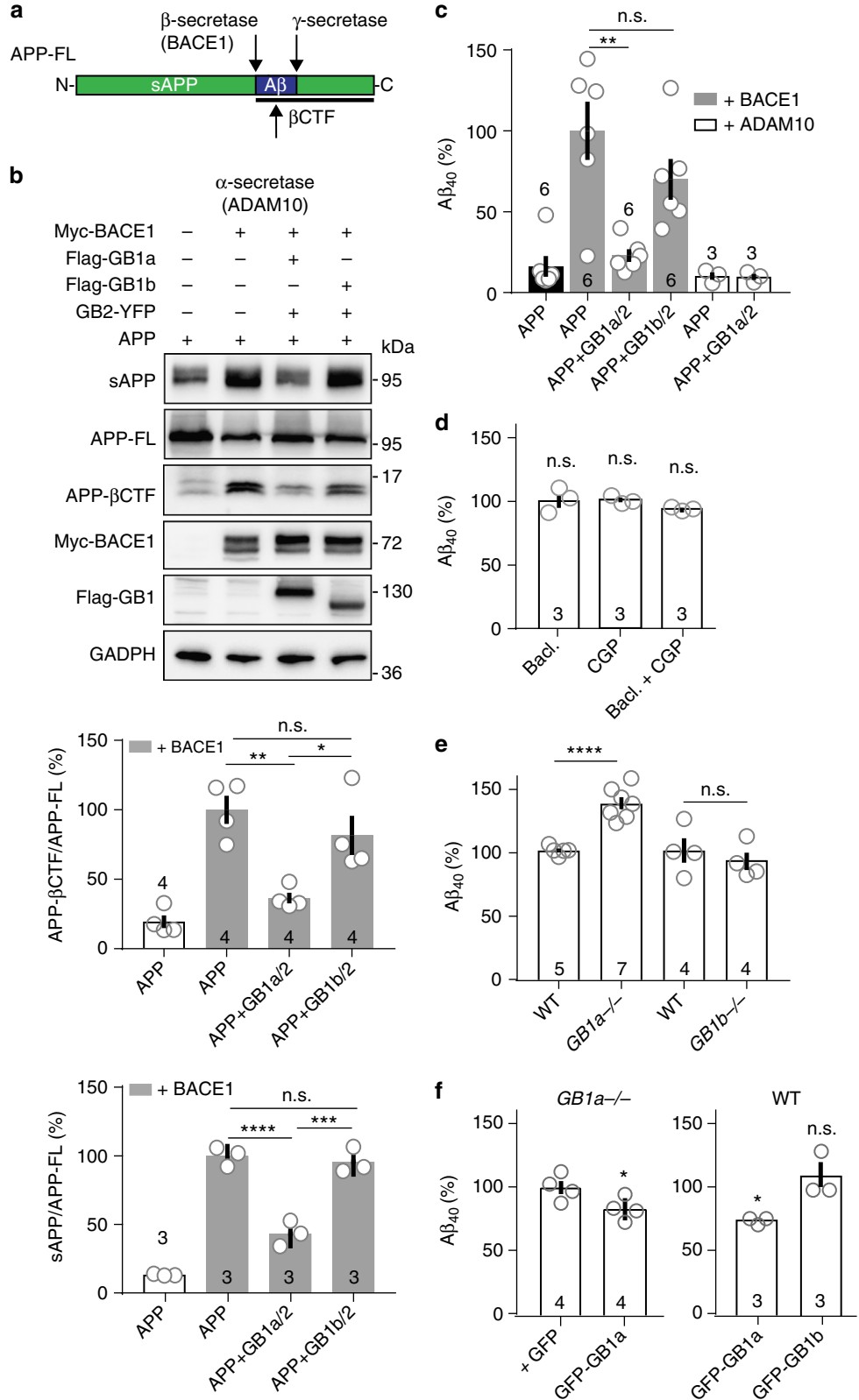

fluorescence at $t_0$; Fig.7c), suggesting that transfected BBS-APPmCherry reached a steady-state level at the plasma membrane.

We additionally studied APP endocytosis in live HEK293 cells using time-lapse confocal microscopy, which allowed monitoring surface BTX-488/APPmCherry internalization (Fig. 7d). In the absence of GBRs or in the presence of GB1b/2 receptors, we observed internalized BTX-488-labeled vesicles after 10 min (Fig. 7d). However, in the presence of GB1a/2 receptors we did not detect internalized BTX-488-labeled vesicles before 30 min.

**Fig. 6** GB1a inhibits BACE1-mediated APP proteolysis and Aβ40 generation. **a** Scheme indicating proteolytic cleavage sites in APP for α-secretase, β-secretase (BACE1) and γ-secretase. APP-FL, APP full-length; sAPP, soluble APP; APP-βCTF, β carboxy-terminal fragment of APP. **b** Immunoblot of APP cleavage products in HEK293 cells expressing Myc-BACE1, APP and GB2-YFP together with Flag-GB1a or Flag-GB1b. For sAPP analysis, the cell culture medium was filtered and concentrated 32 h post-transfection. Glyceraldehyde 3-phosphate dehydrogenase (GADPH) served as loading control. A significant reduction in the APP-βCTF/APP-FL and the sAPP/APP-FL ratio is observed in the presence of GB1a vs. GB1b. One-way ANOVA, $n = 3$–4 independent experiments. **c** Bar graphs of Aβ40 secretion into the culture medium of HEK293 cells expressing APP with or without Myc-BACE1 or Myc-ADAM10 in the presence of GB1a/2 or GB1b/2 (32 h post-transfection). Note that selectively GB1a/2 significantly prevents Aβ40 secretion. One-way ANOVA, $n = 6$ independent experiments. **d** GBR activity does not influence Aβ40 production. Bar graphs of the amount of Aβ40 secreted into the culture medium of WT hippocampal neurons after treatment for 7 days with baclofen (10 μM) or CGP54626 (CGP, 10 nM) or both. Values normalized to untreated (100%). One-way ANOVA, $n = 3$ independent neuronal cultures. **e** Bar graphs of the amount of Aβ40 secreted within 10 days into the culture medium of hippocampal neurons of $GB1a^{-/-}$, $GB1b^{-/-}$ and control WT littermate mice. Neurons from $GB1a^{-/-}$ but not $GB1b^{-/-}$ mice exhibit increased Aβ40 secretion. Unpaired Student's $t$-test, $n = 4$–7 independent neuron preparations. **f** Lentiviral expression of GB1a but not GB1b decreases Aβ40 secretion in neuronal cultures from $GB1a^{-/-}$ and WT mice. Bar graphs of Aβ40 secreted within 10 days into the culture medium of hippocampal neurons infected with purified lentiviral particles expressing GFP, GFP-GB1a or GFP-GB1b. $GB1a^{-/-}$ cultures; unpaired Student's $t$-test, $n = 4$ independent neuronal cultures. WT cultures normalized to uninfected (100%); one-way ANOVA, $n = 3$ independent neuronal cultures. *$P < 0.05$, **$P < 0.01$, ***$P < 0.001$, ****$P < 0.0001$. Data are presented as mean ± s.e.m. Source data are provided as a Source Data file

Simultaneous monitoring of the decay in BTX-488 surface fluorescence showed that GB1a/2 receptors led to a more than 3-fold increase in the respective time constant (no GBRs: $\tau = 5.6$ min, GB1b/2: 5.4 min, GB1a/2: 16.5 min; Fig. 7e). These results indicate that GB1a/2 receptors increase APP surface expression by reducing APP internalization, which prevents amyloidogenic processing in recycling endosomes. GB1a/2 receptors exhibit slower internalization and longer surface stability than GB1b/2 receptors in neurons[11]. It is therefore possible that APP reciprocally stabilizes GB1a/2 receptors at the cell surface, although we did not directly test this.

We addressed whether the presence of GB1a prevents sorting of APP into endosomes in cultured hippocampal neurons (DIV14) transfected with GB1a-GFP, APPmCherry or both. To identify recycling endosomes we incubated neurons with Alexa-AF647 conjugated transferrin. Co-expression of GB1a-GFP indeed significantly decreased the presence of APP-mCherry in transferrin-AF657 positive endosomes (Supplementary Fig. 13). We further analyzed whether neuronal activity influences APP-mCherry and GB1a-GFP sorting into endosomes. To induce neuronal activity we used an established glycine/bicuculine stimulation protocol[29]. This protocol did not significantly alter endosomal localization of APP-mCherry in the presence and absence of GB1a-GFP (Supplementary Fig. 13).

## Discussion
Presynaptic GBRs regulate neurotransmitter release at most synapses in the brain[1]. Presynaptic GBR expression is itself subject to regulation by neuronal activity[3–5] and frequently impaired in disease[6–10]. The SDs of GB1a are essential for presynaptic localization and membrane stabilization of GBRs[11–14]. Proteomic analysis identified several proteins that selectively and directly interact with presynaptic GB1a/2 receptors, including APP, AJAP-1, and PIANP[15,31,32]. We now found that sequence-related epitopes in these proteins interact with the N-terminal SD1 of GB1a. Electrophysiological analysis of $AJAP-1^{-/-}$, $PIANP^{-/-}$, and $APP^{-/-}$ mice revealed that selectively the absence of APP generates a deficit in GBR-mediated presynaptic inhibition. Proteomic, electrophysiological and trafficking data show that binding of APP to the SD1 serves to sort GB1a/2 receptors into axonally destined vesicles. At the same time, binding to surface GB1a/2 receptors interferes with APP processing to Aβ peptides in recycling endosomes. Our results therefore support that APP/GB1a/2 complex formation simultaneously regulates bioavailability and localization of the partner proteins.

APP is reported to link cargo vesicles via adaptor proteins to axonal kinesin-1 motors[22,28,33]. Our proteomic data support that APP/GB1a complexes bind to adaptor proteins of the JIP and CSTN families. BiFC directly showed that APP/GB1a complexes traffic anterogradely in axons. We additionally observed retrograde trafficking of complexes, presumably mediated by dynein motors[28]. $APP^{-/-}$ mice exhibit a 74% reduction but not a complete absence of GBRs in axons. Likewise, $APP^{-/-}$ mice show an impairment but not a complete loss of GBR-mediated inhibition of glutamate release. Differences in the transport velocities of GB1a-GFP and APPmCherry/GB1a-GFP vesicles further suggest the existence of an APP-independent GB1a transport. Possibly, some GB1a/2 receptors also diffuse laterally in the membrane and accumulate at terminals by binding to SD-interacting proteins[14]. It is unclear whether APP, AJAP-1 and/or PIANP retain GB1a/2 receptors at terminals after delivery. The interacting epitopes of SD1 and APP represent intrinsically disordered regions with dynamically interconverting structures, suggestive of a transient regulatory interaction[34,35]. After anterograde trafficking APP may therefore transfer GB1a/2 receptors to the higher affinity binding-sites of PIANP and AJAP-1. In support of such a scenario, $AJAP-1^{-/-}$ and $PIANP^{-/-}$ mice show a trend towards reduced presynaptic GBR-mediated inhibition in electrophysiological experiments. Moreover, both proteins are expected to localize to synaptic adherens junctions[20] and should therefore be well positioned to anchor GB1a/2 receptors at presynaptic terminals. APP/GB1a complexes are also present in dendritic shafts. This is consistent with axonal proteins not being restricted to axons because endoplasmic reticulum and Golgi apparatus extend into dendrites. However, it is also possible that some APP/GB1a complexes internalize in axons and transcytose to the dendrites, as proposed for APP[36,37].

$GB1a^{-/-}$ and $APP^{-/-}$ mice share several phenotypes, including a deficit in GBR-mediated presynaptic inhibition[12] (this study), increased seizure susceptibility[1,17], deficits in long-term potentiation[12,17,38], cognitive impairments[1,12,38], altered network oscillations[39,40], and circadian locomotor activity[1,38]. This is consistent with the proposal that genetic ablation of genes whose protein products belong to the same functional complex produce similar phenotypes[41]. Some phenotypes of $APP^{-/-}$ mice may also relate to a down-regulation of the $K^+$–$Cl^-$ transporter KCC2 and a resulting decrease in GABA$_A$ receptor inhibition[42]. Interestingly, GBRs and KCC2 are reported to associate with one another and GBR activity reduces KCC2 levels at the cell surface[43]. Loss of GBRs may therefore counteract downregulation of KCC2 in $APP^{-/-}$ mice.

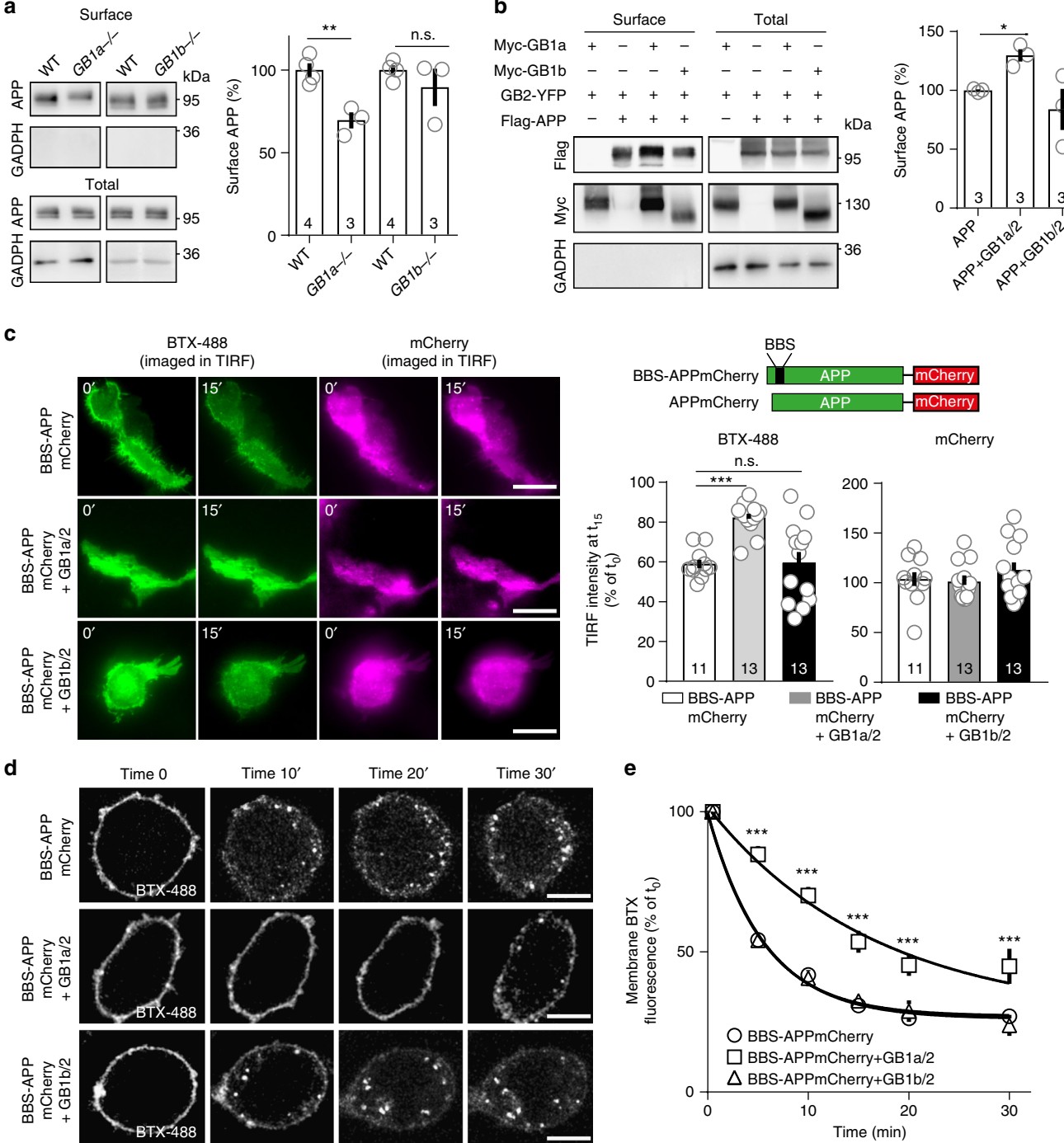

**Fig. 7** GB1a stabilizes APP at the cell surface. **a** Cell surface biotinylation of APP in cultured hippocampal neurons of $GB1a^{-/-}$, $GB1b^{-/-}$, and control WT littermate mice. Bar graph summarizes the densitometric quantification of APP surface levels: WT 100.0 ± 4.1%, $GB1a^{-/-}$ 69.6 ± 4.9%, **$P < 0.01$, unpaired Student's $t$-test; WT 100.0 ± 2.9%, $GB1b^{-/-}$, 89.6 ± 11.2%, $P > 0.05$, Mann-Whitney. **b** Cell surface biotinylation of APP in HEK293 cells in the presence or absence of GB1a or GB1b. Bar graphs summarizes the densitometric quantification of APP surface levels. APP: 100 ± 0.9%; APP + GB1a/2: 129.7 ± 5.3%; APP + GB1b/2: 83.9 ± 17.5%; *$P < 0.05$, one-way ANOVA; $n = 3$ independent experiments. **c** To study APP internalization the α-BTX binding site (BBS) was fused to the extracellular N-terminus of APPmCherry (BBS-APPmCherry). BTX-488 and mCherry cell surface fluorescence of HEK293 expressing BBS-APPmCherry with or without GB1a/2 or GB1b/2 before (time 0') and after BBS-APPmCherry internalization for 15 min at 37 °C (15'). Bar graphs show the mean surface BTX-488 and mCherry fluorescence intensity after 15 min of BBS-APPmCherry internalization. ***$P < 0.001$, one-way ANOVA, BBS-APPmCherry $n = 11$, BBS-APPmCherry + GB1a/2 $n = 13$, BBS-APPmCherry + GB1b/2 $n = 13$ independent experiments. Scale bar 20 μm. **d** Representative confocal images of the BTX-488 fluorescence in HEK293 cells expressing BBS-APPmCherry with or without GB1a/2 or GB1b/2 before (0') and after BBS-APPmCherry internalization for 10, 20, and 30 min. Scale bar 10 μm. **e** Decrease of BTX-488 surface fluorescence in **c** over time. $n = 14$ cells per group, 3 independent transfections per group. ***$P < 0.001$, one-way ANOVA. Data are presented as mean ± s.e.m. Source data are provided as a Source Data file

In amyloidogenic processing, Aβ is liberated from APP by the concerted action of BACE1 and γ-secretase[17,18]. BACE1 is present in axons and dendrites but highly polarized to axonal domains[44], which are the main source of Aβ[45]. Nevertheless, dendritic compartments also release Aβ[46]. BACE1 activity typically occurs in the acidified environment of recycling endosomes[27,47]. While GBR activity influences neither the APP/GB1a interaction nor Aβ40 production, we found that GB1a protects APP from BACE1-dependent endosomal processing by stabilizing APP at the cell surface. Adding to its protective role, GB1a also keeps APP out of dendritic spines that are particularly rich in recycling endosomes[48]. Most neurons in the brain express GB1a, which therefore should markedly influence APP processing. Accordingly, cultured hippocampal neurons of GB1a[−/−] mice exhibited a ~40% increase in secreted Aβ levels compared to WT littermate mice.

While several genome-wide association studies link GBRs to mental health disorders[1,49], no such study directly links GBRs to AD. However, several reports describe a downregulation of GBRs in AD[6,7,49]. GBR downregulation is likely a consequence of the disease, for example caused by increased GBR activity due to excess GABA release by reactive astrocytes[50,51]. Likewise, dysregulated axonal transport, an early pathological feature in AD associated with increased Aβ production[45,52], will reduce the number of GB1a/2 receptors on glutamatergic terminals and promote NMDA receptor-dependent GBR degradation[3,4]. GBR downregulation in AD[6,7] may not only increase Aβ production but also contribute to excitotoxicity and the high incidence in seizures and memory deficits in patients[53], which is supported by the pathology of GB1a[−/−] and APP[−/−] mice. Increased GABA release by reactive astrocytes in AD[50,51] may help to counteract excess glutamate release and therefore play opposing roles in excitotoxic processes.

According to the amyloid hypothesis, accumulation of Aβ in the brain drives AD pathogenesis. Reducing Aβ production is therefore expected to ameliorate AD symptoms[18,19]. Our study shows that stabilizing APP with GB1a at the cell surface prevents Aβ formation. NMDA receptor blockade prevents GBR degradation[3,4] and provides a means to stabilize APP/GB1a complexes. Although controversial, Memantine®, a non-competitive NMDA receptor antagonist used to treat AD patients, is reported to stabilize APP at the cell surface and to reduce Aβ levels[54]. Thus it is possible that Memantine® stabilizes APP at the cell surface by preventing NMDA receptor-induced GBR internalization[3,4]. GBR antagonists provide another means to stabilize GBRs at the cells surface by preventing GBR degradation[51,55]. GBR antagonists are already undergoing evaluation as possible AD therapeutics because they promote excitatory neurotransmitter release and enhance cognition[55]. Moreover, signaling pathways that increase cAMP levels, such as activation of β-adrenergic receptors, increase GBR availability at the cell surface[56]. Thus, pharmacological stabilization of APP/GB1a complexes at the cell surface may have potential for symptomatic amelioration in AD patients.

## Methods

**Molecular biology**. Plasmids were gifts from D. Selkoe and T. Young-Pearse (pCAX-APP695, pCAX-FLAG-APP695, pCAX-APPs-α; Addgene #30137, #30154, #30147), R. Davis (pcDNA3-Flag-JIP-1b, Addgene #52123), R. Derynck (pRK5M-ADAM10, Addgene #31717), W. Almers (NPY-mCherry, Addgene #67156), P. Scheiffele (Synaptophysin-mCherry), M. Di Luca (Myc-BACE1), J.P. Pin (pRK6-Flag-GB1a), and K. Kaupmann (pCI-HA-GB2-YFP). AJAP-1 (Source BioScience) was subcloned into pcDNA3, PIANP cDNA (OriGene) placed into pCMV6 with an HA-tag insertion after amino acid 27. Myc-GB1a, Myc-GB1b, GB1a-GFP and GB1b-GFP, Myc-GB1aΔSD1, Myc-GB1aΔSD2 and Myc-GB1aΔSD1/2 were as described[57]. pRK6-Flag-GB1b was constructed by replacing the GB1a MluI-BamH1 fragment in pRK6-Flag-GB1a with GB1b. GB1a-Rluc, GB1b-Rluc, APP-Venus and APPmCherry were constructed using overlap extension polymerase

chain reaction and cloned into the pCI vector (Promega). For transfection experiments we used the predominant neuronal APP isoform, APP695[17] (hereafter named APP). The APP deletion mutants APPΔGFLD(Δ28–123), APPΔCuBD (Δ124–189), APPΔAcD(Δ191–294) and APPΔCAPPD (Δ295–504) were generated by overlap extension in the pCAX vector. Numbers indicate the residues deleted in APP. To construct APP-VN, the Venus in APP-Venus was replaced with the N-terminal Venus residues 1–172. To construct GB1a-VC or GB1b-VC, the GFP in Myc-GB1a-GFP or Myc-GB1b-GFP was replaced with the C-terminal Venus residues 155–238. Split Venus constructs include the PRARDPPVAT linker 5′ of the Venus fragments. BBS-APPmCherry was created by adding the α-bungarotoxin (BTX) binding site (BBS) WRYYESSLEPYPD at the N-term of APPCherry between amino acids A30 and E31 (Trenzyme, Germany)[27].

**Mouse strains**. GB1a[−/−], GB1b[−/−], and GB2[−/−] mice were kept in the BALB/c background[12,58], APP[−/−] mice in the C57BL/6 background[59]. AJAP-1[−/−] mice were generated by blastocyst injection of embryonic stem cell clone HEPD0583_2_B09 harboring a knockout-first promotor-driven tm1a allele (AJAP-1tm1a(EUCOMM)Hmgu)[60] and subsequent crossing of founders with the Cre-deleter strain B6.C-Tg(CMV-cre)1Cgn/J. In the converted tm1b allele exon 2 of the AJAP-1 gene was deleted leaving a LacZ reporter behind, which contains an en-2 splice acceptor and an internal ribosomal entry site. PIANP[−/−] mice (B6-Pian-p[em1Bet]) in the C57BL/6 background were generated using the Alt-R CRISPR/Cas9 targeting system (IDT, Leuven, Belgium). The Cas9 target sequence 5′-GACCCCA CACTATAGCCCAAGGG-3′ in the Pianp gene (MGI:2441908) was selected using the CRISPOR search algorithm http://crispor.tefor.net. Enzymatic mutation altered the targeting sequence to 5′-GACCCCACACTATAGGTGTGAGATGGG-3′ resulting in a frame shift after P97 and premature termination of translation. All mouse experiments were conducted in accordance with Swiss guidelines and received ethical approval from the veterinary office of Basel-Stadt.

**Affinity purifications from brain membranes**. Plasma-membrane enriched protein fractions were prepared from whole brain isolated from a pool of 10 WT and 2–4 knock-out mice[61]. Concentrations of protein fractions were determined by Bradford assays (Biorad). Membrane proteins were solubilised with CL-47 and CL-91 buffers at 1 mg protein per ml (Logopharm GmbH, Germany). After 30 min incubation on ice and clearing by ultracentrifugation (10 min, 150,000 × g) solubilisates were incubated with the immobilized antibodies and incubated for 2 h on ice. 10–15 μg of the following antibodies were used for an immunoprecipitation out of 1 mg of solubilised membrane proteins: anti-APP, Ab#1, rabbit anti-APP (A8717, Sigma), Ab#2, rabbit anti-APP (ABIN1741750, Antikörper-online), Ab#3, goat anti-APP (sc-7498, Santa Cruz); anti-AJAP-1, Ab#1, sheep anti-AJAP-1 (AF7970, R&D Systems), Ab#2, rabbit anti-AJAP-1 (HPA012157, Sigma), Ab#3, goat anti-AJAP-1 (sc-163371, Santa Cruz); anti-PIANP, Ab#1, rabbit anti-PIANP (PAB21925, Abnova), Ab#2, rabbit anti-PIANP (raised against epitope: mouse PIANP aa 221–237, generated by Young in Frontier, South Korea) (Fig. 1c). For a quantitative comparison of GBRs in two samples (Fig. 1a), a mixture of anti-GB antibodies including rabbit anti-GB1 (322102, Synaptic Systems), rabbit anti-GB2 (322203, Synaptic Systems), guinea pig anti-GB2 (322204/5, Synaptic Systems)) was applied to isolate the complete pool of receptor protein complexes, which was controlled by immunoblot analysis of the respective supernatant after antibody incubation. After two washes, proteins were eluted and the majority processed for MS-analysis. Proteins were separated on SDS-PAGE gels and silver-stained. Lanes were cut into two sections (high and low MW) and digested with sequencing-grade modified trypsin (Promega, Mannheim, Germany). Peptides were extracted and prepared for MS analysis as described[15]. Influence of GBR activity on complex formation was analyzed by incubating unsolubilized membranes with 1 mM GABA or 4 μM CGP54626 in PBS buffer for 1 h at room temperature. Subsequently membrane proteins were solubilised and processed for immunoprecipitations as described above (Supplementary Fig. 2a).

**Mass-spectrometry and protein identification**. Mass spectrometric analysis was carried out as described[61]. Peptide samples dissolved in 0.5% trifluoroacetic acid were loaded onto a trap column (C18 PepMap100, 5 μm particles; Thermo Scientific), separated by reversed phase chromatography via a 10 cm C18 column (PicoTip™ Emitter, 75 μm, tip: 8 μm, New Objective, self-packed with ReproSil-Pur 120 ODS-3, 3 μm, Dr. Maisch HPLC; flow rate 300 nl/min) using an UltiMate 3000 RSLCnano HPLC system (Thermo Scientific), and eluted by an aqueous organic gradient (eluent "A": 0.5% acetic acid; eluent "B" 0.5% acetic acid in 80% acetonitrile; "A"/"B" gradient: 5 min 3% B, 60 min from 3% B to 30% B, 15 min from 30% B to 100% B, 5 min 100% B, 5 min from 100% B to 3% B, 15 min 3% B). Sensitive and high-resolution MS-analyses were executed on an Orbitrap Elite mass spectrometer with a Nanospray Flex Ion Source (both Thermo Scientific). Precursor signals (LC-MS) were acquired with a target value of 1,000,000 and a nominal resolution of 240,000 (FWHM) at m/z 400; scan range 370 to 1700 m/z). LC-MS/MS data were extracted using "msconvert.exe" (part of ProteoWizard; http://proteowizard.sourceforge.net/, version 3.0.6906). Peak lists were searched against a UniProtKB/Swiss-Prot database (containing all rat, mouse and human entries) using Mascot 2.6.0 (Matrix Science, UK). One missed trypsin cleavage and

common variable modifications including S/T/Y phosphorylation were accepted for peptide identification. Significance threshold was set to $p < 0.05$.

**Mass-spectrometry based protein quantification.** Label-free quantification of proteins was based on peak volumes (PVs = peptide $m/z$ signal intensities integrated over time) of peptide features[61]. Peptide feature extraction was done with MaxQuant[62] (http://www.maxquant.org/, version 1.4) with integrated effective mass calibration. Features were then aligned between different LC-MS/MS runs and assigned to peptides with retention time tolerance ±1 min and mass tolerance: ±1.5 ppm using an in-house developed software. The resulting peptide PV tables formed the basis for protein quantification (Fig. 1). For relative quantification of proteins in two samples (Fig. 1a, Supplementary Fig. 2a), protein ratios (rPVs) were determined from protein profiles[61]. Briefly, for each peptide, the PVs were then normalized to their maximum over all AP data sets yielding relative peptide profiles, ranked for each protein by pairwise Pearson correlation. These values were normalized to the molecular abundance of GBRs (Source Data File) to obtain the degree of association with the target. The median from all peptides, assigned as unique for each individual protein, was used to calculate the abundance difference of GBR constituents in knock-outs vs. WT (Fig. 1a). Interactome analysis were performed by comparing relative protein abundance in a sample vs. control (rPV, Fig. 1c, Source Data File), determined by the TopCorr method as the median of at least 2–6 individual peptide PV ratios of the best correlating protein-specific peptides (as determined by Pearson correlation of their abundance values)[63]. Specificity thresholds of APs were determined from rPV histograms of all proteins detected in the respective AP vs. control. Constituents of the GBR proteome were considered specifically co-purified when rPVs (wild-type mouse vs. IgG and KO) were above the threshold.

**Transferrin treatment and neuronal activity induction.** Lipofectamine 2000 (Life Technologies) was used to transfect HEK293 cells. The amount of DNA in the transfections was kept constant by supplementing with pCI plasmid DNA (Promega). For preparation of cultured neurons embryonic day 16.5 mouse hippocampi were dissected, digested with 0.25% trypsin (Invitrogen) in HBSS (Gibco, 14170–088) medium for 13 min at 37 °C, dissociated by trituration and plated on glass coverslips coated with 1 mg/ml poly-L-lysine hydrobromide (PLL, Sigma) in 0.1 m borate buffer (boric acid/sodium tetraborate)[13]. Neurons were seeded at a density of ~550 cells/mm² in neurobasal medium (Invitrogen) supplemented with B27 (Invitrogen) and 0.5 mM L-glutamine and cultured in a humidified atmosphere (5% $CO_2$) at 37 °C. Cultured hippocampal neurons were transfected using Lipofectamine 3000 (Life Technologies) or Effectene (Qiagene). Transferrin-AF647 (Invitrogen) and bicuculline/glycine treatment[29] was performed at DIV14, 30 min before transferrin-AF647 treatment, neurons were incubated with fresh Neurobasal medium. Incubation of Transferrin-AF647 was added at a final concentration of 50 μg/μl for 1 h, For activity induction neurons were treated with 20 μM bicuculline/200 μM glycine for 5 min in Neurobasal medium. The medium was then replaced with fresh medium supplemented with 20 μM bicuculline for 15 min. Control cultures were kept in pure Neurobasal medium. Cultures were washed with 1× PBS, fixed for 10 min at room temperature in 4% PFA/4% sucrose and mounted on microscope slides with Dako Fluorescence Mounting Medium (Agilent). Samples were imaged on Zeiss LSM880 confocal microscope equipped with Plan-Apochromat ×63/1.4 Oil objective. Collected Z-stacks were quantified using Fiji and Adobe Photoshop CC 2018.

**IP and immunoblot experiments.** Hippocampal neurons or HEK293 cells were harvested 24 h after transfection, washed twice with ice-cold PBS, and subsequently lysed in NETN buffer supplemented with complete EDTA-free protease inhibitor mixture (Roche). After rotation for 10 min at 4 °C, lysates were cleared by centrifugation at 10,000×g for 10 min at 4 °C. Lysates were directly used for immunoblot analysis or immunoprecipitated with antibodies coupled to Protein-G/A Agarose beads (Roche). Membrane protein fractions from brains (prepared as described above), lysates from cultured cells and immunoprecipitates were resolved using standard SDS-PAGE gels and decorated with the indicated antibodies. SuperSignal Femto chemiluminescence detection kit (Thermo Scientific) or ECL Prime (Amersham Biosciences) were used for visualization using a Fusion FX Chemiluminescence System (Vilber Lourmat, Witec AG). Band intensities were quantified by ImageJ software (NIH). Uncropped and unprocessed scans of immunoblots are shown in the Source Data file.

To determine whether GBR activity regulates binding of APP, AJAP-1, and PIANP to GB1a, we prepared brain membrane fragments as for [35S]GTPγS binding assays and resuspended them in NET buffer (100 mM NaCl, 1 mM EDTA, 20 mM Tris/HCl, pH 7.4) supplemented with EDTA-free protease inhibitor mixture (Roche) for 90 min at 4 °C. Membranes were treated with 1 mM GABA or 4 μM CGP54626 or left untreated for 1 h at room temperature. Nonidet P-40 detergent was added to a final concentration of 0.5%. Incubation with antibodies (α-APP, (Y188, Abcam), α-AJAP1 (AF7970, R&D Systems), α-PIANP (PAB21925, Abnova)) was for 16 h at 4 °C, followed by IP. For densitometric analysis of immunoblots, the GB1a signal was divided by the signal of the immunoprecipitated protein (APP, AJAP-1, PIANP) and normalized as 1.0 for untreated control samples.

The antibodies used for immunoprecipitation were: rabbit anti-c-myc (C3956, Sigma), mouse anti-c-myc 9E10 (sc-40, Santa Cruz), mouse anti-flag M2 (F1804, Sigma), rabbit anti-flag (F7425, Sigma), rabbit anti-GB1 (rat aa 857–960), rabbit anti-GB2 (322203, Synaptic Systems), rabbit anti-APP Y188 (ab32136, Abcam) and rabbit anti-APP Y188 (ab32136, Abcam). The primary antibodies used for immunoblot analysis were: mouse anti-GB1 (ab55051, Abcam), rabbit anti-GB1 (rat aa 857–960), mouse anti-GB2 (75–124, NeuroMab), mouse anti-c-myc 9E10 (sc-40, Santa Cruz), rabbit anti-c-myc (C3956, Sigma), rabbit anti-flag (F7425, Sigma), mouse anti-APP A4 22C11 (mab348, Millipore), rabbit anti-APP Y188 (ab32136, Abcam), sheep anti-AJAP-1 (AF7970, R&D Systems), rabbit anti-PIANP (PAB21925, Abnova), rabbit anti-Calnexin (ab75801, Abcam), rabbit anti-β-Actin 13E5 (#4970, Cell Signaling) and mouse anti-GADPH (sc-32233, Santa Cruz). The secondary antibodies were: HRP-conjugated anti-rabbit (NA9340V, GE Healthcare, UK), anti-mouse (NA9310V, GE Healthcare, UK), anti-sheep (ab7111, Abcam), anti-mouse (sc-2005, Santa Cruz), anti-rabbit (sc-2004, Santa Cruz).

**Purification of proteins for structural analysis.** SD1/2 was produced as secreted protein in *Sf21* insect cells and subsequently purified by Ni-chelate affinity-matrix and size exclusion chromatography, as described[15]. APP(191–294), AJAP-1 (175–279), and PIANP(27–174) were subcloned into pET30 (Novagen) with a N-terminal His-tag followed by the B1 domain of streptococcal protein G as a solubility enhancement tag (SET) and a TEV cleavage site. Protein expression was induced in *E.coli* BL21 (DE3) by 1 mM IPTG. Cells were either grown on LB-medium (MP Biomedicals) or for isotopic protein labeling on M9 minimal medium made with $^{15}NH_4Cl$ and $^{13}C$-Glucose (both Cambridge Isotope Laboratories). After cultivation (LB medium: 4 h, 37 °C, M9 medium: 16 h, 30 °C) cells were lysed in 20 mM Tris/HCl pH 8, 300 mM NaCl, 10 mM imidazole, 0.5 mM EDTA/EGTA by sonication. Lysates were cleared by centrifugation (20,000 x g, 4 °C, 20 min), loaded on a HisTrap HP sepharose column (GE Healthcare) and His-tagged proteins subsequently eluted with 500 mM imidazole. Respective fractions of the APP construct were pooled and dialyzed against 50 mM Tris/HCl pH 8, 150 mM NaCl, 0.5 mM EDTA, 1 mM DTT before adding the TEV-protease. After 14 h incubation, the digest was loaded on a HisTrap HP sepharose column to remove all His-tag-containing species. The cleavage step was omitted for the AJAP-1 and PIANP constructs. Final purification was done by size exclusion separations (Superdex 200 10/300 GL column, GE Healthcare). The purity of the samples was determined by separating proteins on SDS-PAGE and visualized by conventional Coomassie stain solutions. Proteins were concentrated by Vivaspin® 6 centrifugal concentrators (Vivascience) and directly used for biophysical characterizations.

**Protein NMR spectroscopy.** The NMR spectra were recorded at 293 K on a Bruker Avance 600 equipped with a cryogenically cooled pulsed-field gradient triple-resonance probe (TXI) operating at 600.13 MHz. The sequence-specific assignment of backbone atoms of APP191–294 both in the absence and the presence of SD1/2 protein was obtained from the following experiments: $^1$H-$^{15}$N HSQC, HNCA, HN(CO)CA, HNCO, HN(CA)CO, CBCA(CO)NH, HBHA(CO) NH, CBCANH, NOESY-$^1$H-$^{15}$N-HSQC (250 ms mixing time), and HN(CA)NNH. The interaction site of APP with SD1/2 was determined by observation of chemical-shift changes and cross-peak intensity changes in $^1$H–$^{15}$N HSQC spectra of $^{15}$N-labeled or $^{13}$C/$^{15}$N-labeled APP during titrations with unlabeled SD1/2 protein up to a stoichiometric ratio of 1:1.1. The NMR samples contained 0.1–0.65 mM APP in 50 mM sodium phosphate buffer (pH 6.8) with 50 mM NaCl, 0.5 mM EDTA, 0.5 mM EGTA, and 10% $D_2O$ (v/v). Similarly, complex formation of $^{15}$N-labeled AJAP-1(175–279) or PIANP(27–174), each with a solubility enhancement tag (SET[64], see Protein Expression), was monitored by $^1$H–$^{15}$N HSQC spectra during titrations with unlabeled SD1/2. 4,4-Dimethyl-4-silapentane-1-sulfonate (DSS) was used as internal standard for $^1$H chemical shift referencing.

**Cell surface binding.** To estimate binding of purified Myc-SDI/II to APP, AJAP-1 and PIANP, cDNAs were transfected into tsA201 cells. After 2 days of cultivation, cells were incubated for 30 min at room temperature with SD1/2 in reduced serum OptiMEM® medium. Myc-SD1/2 was added to the medium to give a final concentration of 0.2 to 2000 nM. Cells were subsequently fixed in 4% PFA, blocked in 1% BSA and then incubated with mouse anti-c-Myc 9E10 (11667149001, Roche) to determine bound SD1/2 and goat anti-APP (sc-7498, Santa Cruz), sheep anti-AJAP-1 (AF7970, R&D Systems) or rat anti-HA (HA-PIANP) (clone 3F10, 11867423001, Roche), respectively, and detected with Alexa-conjugated secondary antibodies (Alexa Fluor® 488 donkey anti-goat IgG, Alexa Fluor® 488 donkey anti-rat IgG, Alexa Fluor® 555 donkey anti-mouse IgG all from Life Technologies). Average intensity values of bound and expressed proteins in individual cells using drawn region of interests around the perimeter of each cell were determined after background subtraction and put into ratio. For each measurement, $n = 3$–20 cells were used. Apparent dissociation constants ($K_D$ values) were determined using a Hill equation with a coefficient of 1.

**Electrophysiology.** Three hundred μm-thick hippocampal slices were prepared with a Leica VT1200S vibratome from P12-P21 $APP^{-/-}$, $AJAP$-$1^{-/-}$, $PIANP^{-/-}$ or WT littermate mice and incubated for 15 min at 32 °C in aCSF containing 126 mM NaCl, 26 mM NaHCO₃, 2.5 mM KCl, 1.25 mM NaH₂PO₄, 2 mM CaCl₂, 1 mM

MgCl$_2$, and 10 mM glucose. Slices were kept at room temperature until recording at 32 °C submerged in a recording chamber perfused with ACSF. CA1 pyramidal cells were visually identified using a ×60 objective under video infrared Nomarski optics with a BX51WI microscope (Olympus). Cell cultures were prepared from WT, *APP*$^{-/-}$, or *GB1a*$^{-/-}$ mice as described above. After 10–13 days in vitro, coverslips were transferred into a submerged chamber and perfused with aCSF at 32 °C. Whole-cell voltage-clamp recordings were obtained with 4–6 MΩ borosilicate glass pipettes via an intracellular solution containing 135 mM CsCH$_3$O$_3$S, 8 mM NaCl, 4 mM Mg-ATP, 0.3 mM Na$_3$-GTP, 0.1 mM TEA-Cl, 5 mM QX-314 and 10 mM HEPES. A liquid junctional potential of −10 mV was left uncorrected. Cells were voltage-clamped at −60 mV with a Multiclamp700B amplifier (Molecular Devices). Spontaneous mEPSCs were recorded in the presence of 0.2 μM TTX and 100 μM picrotoxin. EPSCs were evoked with extracellular monopolar current pulses generated by a custom-made isolated current stimulator and applied via a patch-pipette filled with aCSF and positioned to activate the Schaeffer Collateral pathway. All recordings were filtered at 4–10 kHz and digitized at 10–20 kHz with a Digidata 1550B digitizer (Molecular Devices).

**BRET and [$^{35}$S]GTPγS binding assays**. BRET experiments monitoring G protein activation were conducted and analyzed as described[24]. Mouse brain membranes for [$^{35}$S]GTPγS binding assays were prepared and analyzed as described[12].

**Biotinylation assay**. HEK293 cells were biotinylated using the Pierce Cell Surface Protein Isolation Kit (Pierce, 89881). Transfected HEK293 cells on 6-well plates were incubated with 1 mg/ml sulfo-NHS-SS-biotin (Pierce) in PBS for 30 min at 4 °C. After quenching the biotinylation reaction with 50 mM glycine and rinsing of the cells with ice-cold TBS (Tris-buffered saline) and PBS, cells were scrapped from the plates and lysed in NETN buffer (100 mM NaCl, 1 mM EDTA, 0.5% Nonidet P-40, 20 mM Tris/HCl, pH 7.4) supplemented with an EDTA-free protease inhibitor mixture (Roche). The lysate was incubated at 4 °C for 15 min then centrifuged 10,000 × $g$ for 10 min at 4 °C and proteins in the supernatant quantified. Biotinylated surface proteins were purified using NeutrAvidin-agarose (Pierce), washed, and resuspended in protein loading buffer. Proteins were identified on immunoblots.

**Immunofluorescence**. Neurons on glass coverslips were fixed for 5 min in 4% PFA/4% sucrose at RT, permeabilized with PBS$^{+/+}$ (D8662, Sigma, supplemented with 1 mM MgCl2 and 0.1 mM CaCl$_2$))/Triton-0.1%, blocked with PBS$^{+/+}$/5% BSA and labeled with primary antibodies in PBS$^{+/+}$ (D8662, Sigma) and 5% BSA for 2 h and secondary antibodies for 45 min. PBS$^{+/+}$ washes were performed after each antibody incubation. Coverslips were mounted on glass slides in Fluoromount$^{TM}$ (F4680, Sigma). Images were captured using Zeiss LSM-700 system with a Plan-Apochromat 63 × /NA 1.40 oil DIC, using Zen 2010 software.

Primary antibodies used: mouse anti-GB1 (ab55051, Abcam), chicken anti-map2 (ab5392, Abcam), mouse anti-c-myc 9E10 (11667149001, Roche), mouse anti-piccolo (142111, Synaptic System), rabbit anti-KLC1 (ab187179, Abcam), rabbit anti-GFP (ab290, Abcam), mouse anti-flag M2 (F1804, Sigma), mouse anti-PSD-95 (ab2723, Abcam). Secondary antibodies used: Alexa Fluor® 647 donkey anti-chicken IgY (Millipore), Alexa Fluor® 555 donkey anti-mouse IgG (Life Technologies), Alexa Fluor® 488 donkey anti-rabbit IgG (Life Technologies), Alexa Fluor® 647 donkey anti-mouse IgG (Invitrogen), and Alexa Fluor® 568 donkey anti-rabbit IgG (Invitrogen).

**TIRF microscopy and live confocal imaging**. Transfected HEK293 cells were incubated for 15 min at 16 °C in the dark in PBS$^{+/+}$ containing 3 μg/ml BTX conjugated to Alexa-488 (Thermo Scientific). Cells were washed three times with PBS$^{+/+}$ at 16 °C and mounted on 37 °C incubator stages of a TIRF Olympus IX81 inverted Microscope equipped with a motorized TIRF system and an Apo N 60 × / NA 1.49 TIRF objective (Olympus, Japan). Excitation of GFP/Venus and mCherry was at 488 nm and 561 nm, respectively. Images were acquired with a Hamamatsu imagEM c9100–13 EMDCCD camera using Xcellence software (Olympus). TIRF measurements with transfected HEK293 cells was on 35 mm μ-Dishes, high Glass Bottom (Ibidi, Germany), in serum free DMEM/F-12 medium (Gibco 11320–074) (Fig. 7c and Supplementary Fig. 6b and 7b). Live confocal imaging was with a Zeiss point scanning confocal LSM-800 inverted microscope, using a heated stage and a 63 × /NA 1.4 Plan-Apochromat objective. Excitation was at 488 nm and 555 nm; images were collected at a rate of 1 frame/s (Fig. 7d, Supplementary Fig. 10a, Supplementary Movie 1).

**Image analysis and quantification**. Images were taken under identical acquisition parameters for all conditions within the experiment. Saturation was avoided by using image acquisition software to monitor intensity values. Images were analyzed by Fiji analysis software. For quantification, values were averaged over multiple neurons from at least three independent culture preparation. Pearson and Mander correlation coefficient statistics were used to analyze the colocalization between fluorophores using the JACoP plugin of Fiji.

Axon-to-dendrite (A:D) ratio of exogenous Myc-GB1a or Myc-GB1b protein were performed as described[13], using GFP as a volume marker and Fiji analysis

software. For rescue experiments, the neurons were co-transfected with either mCherry, APPmCherry or APLP-2mCherry (Fig. 4e). Kymographs for analysis of vesicle transport were created by drawing one-pixel-wide lines traced from the soma to the axon tip using the KimographBuilder plugin of Fiji. The trafficking velocities were obtained using the Velocity measurement tool. Episodes of directed vesicle movement are represented in kymographs as displacements in the anterograde or retrograde direction. Non-mobile episodes produce straight vertical lines with short horizontal displacements resulting from the "wiggling" of vesicles.

**Aβ40 quantification**. Transfected HEK293 cells were incubated with serum free DMEM/F-12 medium (Gibco 11320–074) for 24 h. After determining the total amount of protein in the supernatant the Wako II Aβ40 ELISA kit was used for Aβ40 quantification. For Aβ40 quantification in neurons, 1 × 10$^5$ neurons from *GB1a*$^{-/-}$, *GB1b*$^{-/-}$ or WT littermate mice were incubated for 10 days in conditioned medium. In some experiments purified lentiviral particles (GeneCopoeia: 217LPP-Rn10234-Lv122 GFP-GB1a, 217LPP-Rn10298-Lv122 GFP-GB1b, 217LPP-EGFP-Lv242 GFP) were added at a concentration of 1 transforming unit per neuron after 3 days for 24 h in 200 μl conditioned medium, before adding 800 μl pre-warmed conditioned medium. After 10 days, the conditioned medium was cleared at 1000×$g$ for 15 min at 4 °C and processed for Aβ40 quantification.

**Statistical analysis**. Data analysis was with GraphPad Prism version 7.0 (GraphPad Software, La Jolla, CA). Individual data sets were tested for normality with the Shapiro-Wilk or the D'Agostino-Pearson test (for $n \geq 8$). Statistical significance of differences between two groups was assessed by unpaired two-tailed Student's $t$-test or ANOVA as indicated. For non-normal distribution the non-parametric Mann–Whitney test was used. $P$-values < 0.05 were considered significant. Data are presented as mean ± standard error of the mean (s.e.m.).

**Reporting summary**. Further information on experimental design is available in the Nature Research Reporting Summary linked to this article.

## Data availability

Data supporting the findings of this study are available within the paper and in the Supplementary Information and Source Data files. The mass spectrometry proteomics data have been deposited to the ProteomeXchange Consortium via the PRIDE partner repository with the dataset identifier PXD012487.

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

## Acknowledgements

The EUCOMM "Tools for Functional Annotation of the Mouse Genome" project (FP7-HEALTH-F4-2010-261492) provided the embryonic stem cell clone HEPD0583_2_B09 used to generate *AJAP-1$^{-/-}$* mice. *PIANP$^{-/-}$* mice were generated in collaboration with P. Pelczar at the Center for Transgenic Models at the University of Basel, Switzerland. This work was supported by grants of the Swiss Science Foundation (31003A-172881 to B.B.), the National Center for Competences in Research (NCCR) 'Synapsy, Synaptic Basis of Mental Health Disease' (to B.B), the Deutsche Forschungsgemeinschaft (SFB 746 to B. F.) and the Czech Academy of Sciences (GACR 16–17823S to R.T.).

## Author contributions

M.C.D., A.R., M.G., B.F., J.S., and B.B. designed research; M.C.D., A.R., A.S., P.D.R., M.S., S.F., T.F., T.L., R.T., M.C., V.B., W.B., D.B., and M.G. performed experiments; M.S. contributed important reagents and M.C.D., A.R., J.S., and B.B. wrote the paper with input from the other authors.

## Additional information

**Competing interests:** The authors declare no competing interests.

**Journal Peer Review Information**: *Nature Communications* thanks Tija Jacob, and other anonymous reviewer(s) for their contribution to the peer review of this work. Peer reviewer reports are available.

