## [Peer Review File · Nature Communications]

Reviewers' comments:

Reviewer #1 (Remarks to the Author):

In the current manuscript using multiple approaches authors examine the trafficking of GABA-receptor to the presynapse and link this trafficking to amyloidogenic APP processing. Authors have identified a GBR/APP multiprotein complex, showed APP dependent GBR localization in the axon, and subsequent APP stabilization that reduces amyloid beta generation. Overall this is an important study as the underlying mechanism of axonal transport of GBR remains unknown. However, some of the major issues remain, such as no data on axonal live-imaging of GBR, and the changes in GBR localization upon neuronal activity induction. As neuronal activity has been shown to modulate APP transport and its subsequent processing, it is imperative to test the same for GBR. This reviewer believes that the majority of the experiments are well-designed, well controlled, and the manuscript is well presented. However, authors should address the issues mentioned above before the manuscript is accepted for publication in Nat Communication. My comments are below:

1. Include the live imaging data on GBR axonal transport.
2. Authors should induce neuronal activity and test GBR localization, and its co-transport with APP.
3. It will be exciting and informative to look at the GBR/APP interaction at the presynapse. Authors have excellent data in the axon shaft, but, they should try to include the synapse data as well.
4. Fig. 5 has some major issues. The representative kymograph has very few mobile APP vesicles, compared to other published reports even when authors took 5 min long movies. Is there any reason to see such a drastic reduction in moving vesicles even at the basal state?
5. In many experiments, proteins were overexpressed for 48 h, which could be very toxic to the neurons inhibiting axonal transport. Authors should try reducing the post-transfection time.
6. Colocalization of GBR with organelle markers is important, and authors should include that data.
7. Scale bar should be added to all the images.

Reviewer #2 (Remarks to the Author):

Dinamarca and colleagues provide a thorough analysis of the interaction between GABA-B receptors (GBR) and APP (among other proteins). This includes APP acting as an adapter protein that links GBR to kinesin motors. Their appears to be a bi-directional relationship here; 1) the interaction with APP plays a major, but not exclusive, role in trafficking GBR to axon and 2) GBR stabilizes APP at the cell surface which reduces APP internalization and reduces A β generation/secretion. Importantly, this is a factor of membrane bound APP as opposed to an extracellular, soluble APP fragment. The studies were elegantly performed and include several excellent controls, such as showing GBR is reduces on the cell surface in APP-null neurons and rescued when APP is expressed in those neurons.

The manuscript provides a cohesive and convincing story of the relationship between GBR and APP. Several minor points should also be considered:

- 1) How does GBR activity (activation or inhibition) alter this interaction? And how does GBR activity alter APP and A β ? This is particularly relevant since GBR is not just a transmembrane protein, but a functional receptor.
- 2) Authors mention that GBR expression is suppressed in AD brain (refs 6,7). The relevance of these changes in AD brain could be further emphasized and highlighted, as well as some mention of whether or not genetic data so far implicates GBR as playing a role in AD.

3) The final paragraph proposes the hypothesis that memantine could lower A β levels. On the surface this hypothesis is intriguing, however the data that memantine has a direct disease-modifying effect is lacking. There are a few papers showing memantine lowers A β , however there is overwhelming evidence that seems to contradict those findings. To give a more rounded conclusion, my suggestion is to mention the memantine hypothesis briefly (if at all since the supporting data is controversial) and instead focus on new ways that GBR could be used to stabilize APP and positively impact A β /AD. This would give the manuscript an end that points to new therapeutic avenues which would appeal to both AD-focused groups and synapse-focused groups.

Reviewer #3 (Remarks to the Author):

Reviewers' comments:

Summary.

In this very interesting, multidisciplinary manuscript, Dinamarca et al. identify APP, AJAP-1 and PIANP as interactors with GABAB1a/2 receptors using a proteomic approach. The authors find that APP, AJAP-1 and PIANP participate in distinct GBR complexes. The APP/GB1a complex also showed association with the kinesin-1 adaptor calyntenin proteins family. After showing nanomolar affinity of epitopes in APP, AJAP-1 and PIANP with the N-terminal sushi-domain SD1 of GABAB1a, they assessed the role of these interactions in presynaptic GABAB receptor(GBR) transport and expression. Analysis of knockout mice shows that loss of APP reduces GBR expression in axons and GABAB-mediated presynaptic inhibition of glutamate release, while PIANP^{-/-} and AJAP-1^{-/-} mice show no deficit. Using BiFC, they identified that APP/GB1a complex formation mediates presynaptic GB1a/2 receptor trafficking to axons and importantly, also the availability of APP for endosomal cleavage to A β . This manuscript is highly likely to influence the field, and has major implications for potential treatment development in AD.

Overall the experiments are rigorously designed and the data are mostly well presented. However, a few straightforward experiments performed in neurons (using the existing reagents) are needed to strengthen some of the conclusions in a revised version- see major comments points 4 and 5 below. Additional measurement from the trafficking vesicles is also needed: point 3. Minor comments follow, indicating missing scale bars, and typos.

Note: Interestingly, a published abstract from an oral presentation (Rice et al'2017 - [https://www.alzheimersanddementia.com/article/S1552-5260\(17\)32991-6/fulltext](https://www.alzheimersanddementia.com/article/S1552-5260(17)32991-6/fulltext)) also used proteomics to identify an interaction between APP and the sushi domain of the GABAB1a subunit. This abstract describes how sAPP reduces both the frequency of mEPSCs and synaptic vesicle exocytosis in hippocampal cultures. The effect of sAPP on synaptic release is blocked by pretreatment with a GABA(B)R antagonist, demonstrating that sAPP decreases the probability of release via presynaptic GABA(B) receptors. A full length manuscript has not been published yet, but may be under review.

Major comments

1. Raw data from mass spec or blots showing the co-IP(Fig 1c) should be provided in the manuscript or as supplementary data.
2. Page 5 line 4: The authors should include some discussion of why there is no functional change in ARAP1 & PIANP KO mice in terms of GBR-mediated presynaptic inhibition. I assume the authors are analyzing ARAP1 & PIANP KO mice via immunofluorescence/immunohistochemistry to see what disruption of these complexes does to GBR localization (likely another paper in progress).

3. Page 7/Fig 5 (APP/GB1a complexes co-localize with vesicular trafficking proteins in axons).

- The authors should measure the vesicle speed to see if it is consistent with kinesin-1 (forward & backward) rates: KIF1A particles move anterogradely in hippocampal neurons with an average velocity of 1.0 $\mu\text{m/s}$, and retrogradely at approx 0.72 $\mu\text{m/s}$.
- Show some full size images of neurons for CSTN1 & 3, JIP-1b, JIP3 immunofluorescence (also with the KLC1 & VENUS from the APP/GB1) as supplementary data.

4. Page 8, 2nd paragraph (Interaction of APP with GB1a inhibits A β generation): why is there no data presented on quantification of sAPP cleavage product formation, only APP- $\beta\text{CTF}/\text{APP-FL}$ ratio? This should be provided. A critical additional experiment here is to show overexpression of GB1a in neurons can rescue the increased A β 40 secretion measured in hippocampal neurons (addition to Fig 6d). It would be very informative also to know if sAPP β and α are increased in the GB1a -/- neurons, as sAPP β and α have neurotrophic properties and sAPP α knock-in can rescue APP-/- deficits in spatial memory and LTP (Ring et al. 2007).

5. Figure 7: The α -bungarotoxin (BTX) binding experiments to measure endocytosis are regularly performed in neurons, and this should also be done here. GB1A receptors have a much longer surface stability (slower internalization) compared to GB1B in hippocampal neurons as shown by the lab of Trevor Smart (Ref 11, Hannan et al'12). This prior finding should also be mentioned in this publication, as it is highly relevant to the data in Fig 7. In 7E, is the endocytosis rate of GB1a with APP significantly different than GB1b or not – no statistics are provided or significance indicated in the figure or legend.

6. Discussion

- Although the data in this manuscript support the possibility that GBR downregulation in AD may contribute to excitotoxicity and patient memory deficits, evidence of increased GABA release by astrocytes and enhanced GABAB activity in AD has been also proposed (and GABAB antagonists as potential AD therapeutics). These contrasting viewpoints should be mentioned.
- The statement: " We propose that Memantine® stabilizes APP at the cell surface by preventing NMDA receptor-induced GBR internalization " – is too strong of a statement, and implies experiments to test this have been performed in the manuscript. Statement could be " thus it is possible that Memantine® stabilizes APP at the cell surface ..."

Minor comments

- Fig 3. No information given on numbers of cultures used, only "n=10 neurons"
- Fig 5. No information given on numbers of cultures used, only "n=11 neurons"
- Fig 7. no scale bar provided in 7C or D
- Table 1: add genotypes across top of all pages
- APP-/- mice also show reductions in KCC2, a neuron-specific K⁺-Cl⁻ cotransporter and a corresponding depolarizing shift in the GABA reversal potential (EGABA).
- <https://www.ncbi.nlm.nih.gov/pmc/articles/PMC5224924/>

Spelling

P. 8 and Fig 5 legend

anterogradely or retrogradely, should be anterogradely and retrogradely

Fig 6 legend: form, should be from

Fig 7 legend: # of independent transfections not given, only n=14 cell, and

P. 10 lack of AJAP-1-/- and PIANP-/- mice: remove word "lack"

We appreciate the constructive reviews of our manuscript as well as the opportunity to revise our work. The reviewers' comments are reiterated (black), followed by our response (blue) and the revisions made to the manuscript (red).

Please find below our point-by-point reply to reviewer comments.

Reviewer #1

Query 1. Include the live imaging data on GBR axonal transport.

Reply 1. We have collected live imaging data on axonal GB1a-GFP transport in neurons transfected with GB1a-GFP, APPmCherry or both (new Supplementary Fig. 10b). The data show that less GB1a-GFP than APPmCherry or APPmCherry/GB1a-GFP vesicles are mobile. Moreover, we have determined the trafficking speeds of GB1a-GFP, GB1a-GFP/APPmCherry and APPmCherry vesicles, as requested by Reviewer 3. We made the following changes to the manuscript:

P8: The percentage of mobile APPmCherry vesicles in axons was similar to that reported for APP vesicles in cultured hippocampal/cortical neurons¹ and similar to that of APP-VN/GB1a-VC (**Fig. 5g**) or APPmCherry/GB1a-GFP vesicles (**Supplementary Fig. 10b**). However, significantly less GB1a-GFP (23%) than APPmCherry (73%) or APPmCherry/GB1a-GFP (66%) vesicles were mobile (**Supplementary Fig. 10b**), suggesting that endogenous APP is limiting for axonal transport of overexpressed GB1a-GFP. The fraction of vesicles trafficking anterogradely or retrogradely in axons was similar for APP-VN/GB1a-VC and APPmCherry (**Fig. 5g, Supplementary Fig. 10b**). The mean anterograde ($2.2 \pm 0.4 \mu\text{m/s}$, $n = 29$ vesicles) and retrograde ($2.0 \pm 0.2 \mu\text{m/s}$, $n = 29$) trafficking velocities of APPmCherry in axons were similar as reported for dendrites². APPmCherry/GB1a-GFP (antero: $1.6 \pm 0.2 \mu\text{m/sec}$, $n = 16$; retro: $1.3 \pm 0.2 \mu\text{m/sec}$, $n = 26$) and APP-VN/GB1a-VC vesicles (antero: 1.6 ± 0.1 , $n = 19$; retro: $1.5 \pm 0.1 \mu\text{m/sec}$, $n = 8$) had lower mean transport velocities than APPmCherry alone. GB1a-GFP vesicles (anterograde: 1.0 ± 0.1 , $n = 15$; retrograde: $1.2 \pm 0.1 \mu\text{m/sec}$, $n = 25$) had even lower transport velocities, similarly as observed before³.

P11: Differences in the transport velocities of GB1a-GFP and APPmCherry/GB1a-GFP vesicles further suggest the existence of an APP-independent GB1a transport.

P21: Kymographs for analysis of vesicle transport were created by drawing one-pixel-wide lines traced from the soma to the axon tip using the KimographBuilder plugin of Fiji. The trafficking velocities were obtained using the Velocity measurement tool.

Supplementary Fig. 10 (b) Kymographs of vesicles in axons of cultured hippocampal neurons expressing GB1a-GFP, APPmCherry or both. Overlay of kymographs from axons coexpressing GB1a-GFP and APPmCherry identifies APPmCherry/GB1a-GFP complexes. Likewise, time-lapse imaging of axons co-expressing GB1a-GFP and APPmCherry identifies mobile APPmCherry/GB1a-GFP vesicles (arrowheads, acquisition times in seconds). TIRF imaging was 24 hours post-transfection at DIV8. Bar graphs show that fewer GB1a-GFP than APPmCherry or APPmCherry/GB1a-GFP vesicles are mobile. The number of vesicles moving antero- and retrogradely per axons within 5 min are shown in the box plots. Scale bars 25 μ m.

Query 2. Changes in GBR localization upon neuronal activity induction: As neuronal activity has been shown to modulate APP transport and it's subsequent processing, it is imperative to test the same for GBR.

Authors should induce neuronal activity and test GBR localization, and its co-transport with APP.

Reply 2.

Axonal trafficking of GBRs largely takes place in Golgi-derived NPY-positive vesicles. The axonal transport we study in our manuscript is independent of neuronal activity since trafficking experiments in cultured neurons were carried out before synapse-formation takes place.

Sustained activation of glutamate receptors does not influence the kinetics of GBR endocytosis but inhibits recycling of the receptors, resulting in increased sorting to lysosomes and degradation (Benke, D. *Biochem Pharmacol* **86**, 1525-1530,2013). Neuronal activity may therefore affect surface GBR levels and thereby influence sorting of APP/GB1a complexes into early endosomes, in which processing of APP to A β takes place. We have induced neuronal activity in DIV14 neurons transfected with APPmCherry, GB1a-GFP or both and analyzed the routing of transfected proteins into transferrin-AF657 positive endosomes. The data show that co-expression of GB1a-GFP with APPmCherry under baseline conditions indeed significantly reduces APPmCherry co-localization with transferrin-AF657 positive endosomes. However, the protocol used to induce neuronal activity did not significantly alter endosomal localization of APP-mCherry in the presence or absence of exogenous GB1a-GFP. We added these experiments to the revised manuscript.

P10: We addressed whether the presence of GB1a prevents sorting of APP into endosomes in cultured hippocampal neurons (DIV14) transfected with GB1a-GFP, APP-mCherry or both. To identify recycling endosomes we incubated neurons with Alexa-AF647 conjugated transferrin. Co-expression of GB1a-GFP indeed significantly decreased the presence of APP-mCherry in transferrin-AF647 positive endosomes (**Supplementary Fig. 12**). We further analyzed whether neuronal activity influences APP-mCherry and GB1a-GFP sorting into endosomes. To induce neuronal activity we used an established glycine/bicuculline stimulation protocol². This protocol did not significantly alter endosomal localization of APP-mCherry in the presence and absence of GB1a-GFP (**Supplementary Fig. 12**).

P16: Transferrin-AF647 (Invitrogen) and bicuculline/glycine treatment² was performed at DIV14. 30 min before transferrin-AF647 treatment, neurons were incubated with fresh Neurobasal medium. Incubation of Transferrin-AF647 was added at a final concentration of 50 µg/µl for 1 hour, For activity induction neurons were treated with 20 µM bicuculline/200 µM glycine for 5 min in Neurobasal medium. The medium was then replaced with fresh medium supplemented with 20 µM bicuculline for 15 minutes. Control cultures were kept in pure Neurobasal medium. Cultures were washed with 1×PBS, fixed for 10 minutes at room temperature in 4% PFA/4% sucrose and mounted on microscope slides with Dako Fluorescence Mounting Medium (Agilent). Samples were imaged on Zeiss LSM880 confocal microscope equipped with Plan-Apochromat 63×/1.4 Oil objective. Collected Z-stacks were quantified using Fiji and Adobe Photoshop CC 2018.

Supplementary Figure 12 Expression of GB1a-GFP in cultured hippocampal neurons decreases localization of APP-mCherry in early endosomes. Transfected hippocampal neurons were incubated with transferrin-AF647 at DIV14 to determine co-localization of APP-mCherry with transferrin-AF647 positive endosomes. Neuronal activity was elicited in parallel cultures using 20 µM bicuculline and 200 µM glycine. GB1a-GFP significantly decreased the presence of APP-mCherry in transferrin-AF647 positive endosomes. Neuronal activity did not significantly alter endosomal localization of APP-mCherry, both in the presence and absence of GB1a-GFP. Binary images indicate the fraction of APP-mCherry (magenta) present in early endosomes (green) as white regions. Scale bars 10 µm. Bar graphs indicate the ratio of APP-mCherry to total APP-mCherry in early endosomes. Statistical analysis was performed with one-way ANOVA and Tukey's multiple comparison test. $n = 4 - 9$ neurons.

Query 3. It will be exciting and informative to look at the GBR/APP interaction at the presynapse. Authors have excellent data in the axon shaft, but they should try to include the synapse data as well.

Reply 3. We have imaged the APP-VN/GB1a-VC complex in cultured hippocampal neurons co-transfected with the presynaptic marker Synaptophysin-mCherry. We found that the BiFC complex partly co-localizes with Synaptophysin-mCherry and PSD-95. We have included the data into a new Supplementary Fig. 8 and added the following comments to the manuscript.

P7: The APP-VN/GB1a-VC complex also co-localized with the co-expressed presynaptic marker Synaptophysin-mCherry at boutons opposing PSD-95-positive spines (Supplementary Fig. 8).

Supplementary Figure 8 The APP-VN/GB1a-VC complex and GB1a-GFP localize to synaptic boutons. Cultured hippocampal neurons were transfected with either APP-VN/GB1a-VC (BiFC) or GB1a-GFP together with Synaptophysin-mCherry. Neurons were fixed at DIV14 and immunolabeled for PSD-95. Synaptic boutons were identified by Synaptophysin-mCherry fluorescence opposing PSD-95 puncta. Fluorescence for both the APP-VN/GB1a-VC complex and GB1a-GFP were detected at synaptic boutons (arrowheads).

Query 4. Fig. 5 has some major issues. The representative kymograph has very few mobile APP vesicles, compared to other published reports even when authors took 5 min long movies. Is there any reason to see such a drastic reduction in moving vesicles even at the basal state?

Reply 4. The kymograph in Fig. 5f corresponds to the time-lapse images of the anterogradely trafficking APP-VN/GB1a-VC complex shown to the left. We decided to show time-lapse images of a well-separated APP-VN/GB1a-VC complex to make trafficking in the narrow profiles of axons better visible. The kymograph in Fig. 5f is therefore not a representative kymograph. Representative kymographs are shown in the Supplementary Fig. 10a (previously Supplementary Fig. 6). Published data on axonal trafficking in mixed cortical/hippocampal neurons indicate 76% mobile APP vesicles¹, a similar percentage as observed in our experiments (67%, Fig. 5g; 73%, Supplementary Fig. 10b). We visualized axonal transport 12 hours (confocal imaging, Supplementary Fig. 10a) or 24 hours (TIRF imaging, new Supplementary Fig. 10b) after transfection, a time when the fluorescence of tagged proteins was clearly visible. To put our trafficking data in context with literature data and to clarify that Fig. 5f does not show a representative kymograph we have added the following comments to the manuscript:

P8: The percentage of mobile APPmCherry vesicles in axons was similar to that reported for APP vesicles in cultured hippocampal/cortical neurons¹ and similar to that of APP-VN/GB1a-VC (**Fig. 5g**) or APPmCherry/GB1a-GFP vesicles (**Supplementary Fig. 10b**). However, significantly less GB1a-GFP (23%) than APPmCherry (73%) or APPmCherry/GB1a-GFP (66%) vesicles were mobile (**Supplementary Fig. 10b**), suggesting that endogenous APP is limiting for axonal transport of overexpressed GB1a-GFP.

Fig. 5f Time-lapse images of a well-separated APP-VN/GB1a-VC complex trafficking anterogradely in axons (acquisition times in seconds).

Supplementary Figure 10 (b) Kymographs of vesicles in axons of cultured hippocampal neurons expressing GB1a-GFP, APPmCherry or both. Overlay of kymographs from axons coexpressing GB1a-GFP and APPmCherry identifies APPmCherry/GB1a-GFP complexes. Likewise, time-lapse imaging of axons co-expressing GB1a-GFP and APPmCherry identifies mobile APPmCherry/GB1a-GFP vesicles (arrowheads, acquisition times in seconds). TIRF imaging was 24 hours post-transfection at DIV8. Bar graphs show that fewer GB1a-GFP than APPmCherry or APPmCherry/GB1a-GFP vesicles are mobile. The number of vesicles moving antero- and retrogradely per axons within 5 min are shown in the box plots. Scale bars 25 μ m.

Query 5. In many experiments, proteins were overexpressed for 48 h, which could be very toxic to the neurons inhibiting axonal transport. Authors should try reducing the post-transfection time.

Reply 5. For axonal trafficking experiments, we overexpressed transfected proteins for 12-24 hours (Fig. 5f,g; Supplementary Fig. 10a,b). A post-transfection time of 12 (confocal imaging, Supplementary Fig. 10a) to 24 hours (TIRF imaging, Supplementary Fig. 10b) was necessary for confident evaluation of axonal transport. In the previous Supplementary Fig. 6 (now Supplementary Fig. 10a) the post-transfection time was erroneously indicated as 2 days instead of 12 hours. We have corrected this mistake.

Supplementary Fig. 10a Representative kymographs of fluorescent vesicles in axons expressing APPmCherry or APP-VN/GB1a-VC BiFC complexes. Confocal imaging was 12 hours posttransfection at DIV 7 (1 frame/sec).

Query 6. Colocalization of GBR with organelle markers is important, and authors should include that data.

Reply 6. It is well-known that heteromeric GBRs assemble in the endoplasmic reticulum and traffic via the Golgi apparatus to the cell surface (Benke, D. *Biochem Pharmacol* **86**, 1525-1530, 2013), as expected for transmembrane proteins. Consistent with this biosynthetic pathway, we show that APP-VN/GB1a-VC complexes assemble in the perinuclear region and require GB2 for exit from the endoplasmic reticulum (Supplementary Fig. 7b). We now additionally show that NPYmCherry-positive vesicles convey the majority of GB1a-GFP and APP-VN/GB1a-VC protein (new Supplementary Fig. 11). We have added the following statements to the manuscript.

Most relevant in the context of our manuscript is the trafficking of GBRs and APP to recycling endosomes. As outlined above (Query 2) we now show that after endocytosis, GB1a-GFP alone or in combination with APP-mCherry localizes to Tf-positive endosomes, in which amyloidogenesis takes place. Importantly, the data show GB1a-GFP stabilizes APPmCherry at the cell surface, which reduces localization of APP in recycling endosomes (Supplementary Fig. 12).

P7: BiFC imaging in HEK293 cells using total internal reflection fluorescence (TIRF) microscopy showed that the APP-VN/GB1a-VC complex is assembled early in the biosynthetic pathway in the perinuclear region and requires GB2 for surface expression⁴ (Supplementary Fig. 7b).

P8: The majority of APP vesicles colocalized with NPY, a marker for Golgi-derived vesicles². 51% GB1a-GFP and 54% APP-VN/GB1a-VC axonal vesicles also contained NPY-mCherry (Supplementary Fig. 11).

Supplementary Fig. 11 NPY-mCherry positive vesicles convey GB1a-GFP and APP-VN/GB1a-VC protein in axons. Hippocampal neurons were transfected at DIV5 and imaged at DIV6. Arrowheads indicate co-localization of NPY-mCherry with GB1a-GFP or APP-VN/GB1a-VC (BiFC). Scale bars 10 μ m (top), 5 μ m (bottom). Bar graphs indicate the percentage of GB1a-GFP and APP-VN/GB1a-VC positive vesicles that contain NPY-mCherry, as well as the Mander's and Pearson coefficients for co-localization with NPY-mCherry.

Query 7. Scale bar should be added to all the images.

Reply 7. We have added scale bars to all Figures.

Reviewer #2

Query 1. How does GBR activity (activation or inhibition) alter this interaction? And how does GBR activity alter APP and A β ? This is particularly relevant since GBR is not just a transmembrane protein, but a functional receptor.

Reply 1. We analyzed whether GBR activity alters the GB1a/APP interaction. In affinity-purification experiments, GBR activity did not influence APP, AJAP-1 or PIANP binding to the receptor (new Supplementary Fig. 2a,b). Similarly, BRET experiments revealed no BRET changes between GB1a-RLuc and APP-Venus in the presence of GABA or CGP5626 (new Supplementary Fig. 2c,d). Moreover, GBR activity did not influence A β production in cultured neurons (new Fig. 6d).

P4: GBR activity did not significantly influence the amount of APP, AJAP-1, PIANP and other GB1-interacting proteins co-immunoprecipitating with GB1a (**Supplementary Fig. 2a,b**). Likewise, GBR activity did not modify the bioluminescence resonance energy transfer (BRET) between APP-Venus and GB1a-Rluc in transfected HEK293 cells (**Supplementary Fig. 2c,d**).

P9: GBRs activity did not influence A β 40 secretion in cultured hippocampal neurons (**Fig. 6d**), consistent with GBR activity not influencing the APP/GB1a interaction (**Supplementary Fig. 2**).

P12: While GBR activity influences neither the APP/GB1a interaction nor A β 40 production, we found that GB1a protects APP from BACE1-dependent endosomal processing by stabilizing APP at the cell surface.

P14: Influence of GBR activity on complex formation was analysed by incubating unsolubilized membranes with 1 mM GABA or 4 μ M CGP54626 in PBS buffer for 1 hour at room temperature. Subsequently membrane proteins were solubilised and processed for immunoprecipitations as described above (**Supplementary Fig. 2a**).

P17: To determine whether GBR activity regulates binding of APP, AJAP-1, PIANP to GB1a we prepared brain membrane fragments as for [³⁵S]GTP γ S binding assays and resuspended them in NET buffer (100 mM NaCl, 1 mM EDTA, 20 mM Tris/HCl, pH 7.4) supplemented with EDTA-free protease inhibitor mixture (Roche) for 90 min at 4°C. Membranes were treated with 1 mM GABA or 4 μ M CGP54626 or left untreated for 1 hour at room temperature. Nonidet P-40 detergent was added to a final concentration of 0.5%. Incubation with antibodies (α -APP,

(Y188, Abcam), α -AJAP1 (AF7970, R&D Systems), α -PIANP (PAB21925, Abnova)) was for 16 h at 4°C, followed by IP. For densitometric analysis of immunoblots, the GB1a signal was divided by the signal of the immunoprecipitated protein (APP, AJAP-1, PIANP) and normalized as 1.0 for untreated control samples.

Fig. 6 (d) GBR activity does not influence A β 40 production. Bar graphs of the amount of A β 40 secreted into the culture medium of WT hippocampal neurons after chronic treatment for 7 days with baclofen (10 μ M) or CGP54626 (CGP, 10 nM) or both. Values normalized to untreated (100%): Bacl 100.7 \pm 8.6%, CGP 99.7 \pm 1.6%, Bacl. + CGP 94.3 \pm 0.8%, $n = 3$ independent neuronal cultures; $P > 0.05$, one-way ANOVA

Supplementary Fig. 2 GBR activity does not influence the interaction of AJAP-1, PIANP or APP with GB1a. **(a)** Proteomic analysis of native GBR complexes after receptor activation and blockade. Protein abundance ratios for GBR proteome constituents ($n = 4$ measurements) in GBR IPs from membrane fractions pre-incubated with baclofen or CGP54626 (normalized to GB1/2). Proteins were solubilized for IP with the intermediate stringency detergent CL91. GBR activity does not influence binding of GB1a-specific interactors (marked in blue). **(b)** Left: Activation or blockade of GBRs with GABA (1 mM) or CGP54626 (CGP, 4 μ M), respectively, for 1 hour at room temperature in non-solubilized brain membrane fragments did not significantly alter the amount of GB1a protein co-immunoprecipitating with APP, AJAP-1 or PIANP. IPs from untreated membrane fragments served as controls. Right: Bar graphs summarizing the densitometric quantification of co-immunoprecipitated GB1a protein relative to the immunoprecipitated protein. GB1a/APP: control 1.0, GABA 1.18 \pm 0.25, CGP 1.17 \pm 0.46; GB1a/AJAP-1: control 1.0, GABA 1.05 \pm 0.22, CGP 1.04 \pm 0.22; GB1a/PIANP: control 1.0, GABA 1.11 \pm 0.28, CGP 0.93 \pm 0.20 ($P > 0.05$, Tukey's multiple comparison test). $n = 4-5$ mice. **(c)** Scheme depicting that complex formation between APP-Venus and GB1a-RLuc leads to BRET. GB1b-RLuc serves as a negative control. **(d)** GABA (1 mM) and CGP54626 (25 μ M) did not alter BRET between APP-Venus and GB1a-RLuc in HEK293 cells co-transfected with GB2. Representative traces from 3 independent experiments in quadruplicates are shown.

Query 2. Authors mention that GBR expression is suppressed in AD brain (refs 6,7). The relevance of these changes in AD brain could be further emphasized and highlighted, as well as some mention of whether or not genetic data so far implicates GBR as playing a role in AD.

Reply 2. We have added the following information to the Discussion:

P12: While several genome-wide association studies link GBRs to mental health disorders^{4,5}, no such study directly links GBRs to AD. However, several reports describe a downregulation of GBRs in AD⁵⁻⁷. GBR downregulation is likely a consequence of the disease, for example caused by increased GBR activity due to excess GABA release by reactive astrocytes^{8,9}. Likewise, dysregulated axonal transport, an early pathological feature in AD associated with increased A β production^{10,11}, will reduce the number of GB1a/2 receptors on glutamatergic terminals and promote NMDA receptor-dependent GBR degradation^{12,13}. GBR downregulation in AD^{6,7} may not only increase A β production but also contribute to excitotoxicity and the high incidence in seizures and memory deficits in patients¹⁴, which is supported by the pathology of *GB1a*^{-/-} and *APP*^{-/-} mice. Increased GABA release by reactive astrocytes in AD^{8,9} may help to counteract excess glutamate release and therefore play opposing roles in excitotoxic processes.

Query 3. The final paragraph proposes the hypothesis that memantine could lower A β levels. On the surface this hypothesis is intriguing, however the data that memantine has a direct disease-modifying effect is lacking. There are a few papers showing memantine lowers A β , however there is overwhelming evidence that seems to contradict those findings. To give a more rounded conclusion, my suggestion is to mention the memantine hypothesis briefly (if at all since the supporting data is controversial) and instead focus on new ways that GBR could be used to stabilize APP and positively impact A β /AD. This would give the manuscript an end that points to new therapeutic avenues which would appeal to both AD-focused groups and synapse-focused groups.

Reply 3. We have revised the final paragraph to accommodate the reviewer's suggestions.

P12: Although controversial, Memantine®, a non-competitive NMDA receptor antagonist used to treat AD patients, is reported to stabilize APP at the cell surface and to reduce A β levels¹⁵. Thus it is possible that Memantine® stabilizes APP at the cell surface by preventing NMDA receptor-induced GBR internalization^{12,13}. GBR antagonists provide another means to stabilize GBRs at the cells surface by preventing GBR degradation^{9,16}. GBR antagonists are already undergoing evaluation as possible AD therapeutics because they promote excitatory neurotransmitter release and enhance cognition¹⁶. Moreover, signaling pathways that increase cAMP levels, such as activation of β -adrenergic receptors, increase GBR availability at the cell surface¹⁷. Thus, pharmacological stabilization of APP/GB1a complexes at the cell surface may have potential for symptomatic amelioration in AD patients.

Reviewer #3

However, a few straightforward experiments performed in neurons (using the existing reagents) are needed to strengthen some of the conclusions in a revised version- see major comments points 4 and 5 below. Additional measurement from the trafficking vesicles is also needed: point 3.

Note: Interestingly, a published abstract from an oral presentation (Rice et al'2017 - [https://www.alzheimersanddementia.com/article/S1552-5260\(17\)32991-6/fulltext](https://www.alzheimersanddementia.com/article/S1552-5260(17)32991-6/fulltext)) also used proteomics to identify an interaction between APP and the sushi domain of the GABAB1a subunit. This abstract describes how sAPP reduces both the frequency of mEPSCs and synaptic vesicle exocytosis in hippocampal cultures. The effect of sAPP on synaptic release is blocked by pretreatment with a GABA(B)R antagonist, demonstrating that sAPP decreases the probability of release via presynaptic GABA(B) receptors. A full length manuscript has not been published yet, but may be under review.

Reply: Thank you for making us aware of this abstract confirming our initial proteomic finding that GBRs bind to APP. The abstract suggests that sAPP acts as an agonist or positive allosteric modulator of GBRs. In our experiments, we did not observe any agonistic or antagonistic effects of APP or sAPP α on GBR-induced G-protein activation, using BRET and GTP γ S binding to brain membrane preparations (Supplementary Fig. 3). However, our experiments do not rule out that sAPP is an allosteric modulator of GBRs.

Query 1. Raw data from mass spec or blots showing the co-IP(Fig 1c) should be provided in the manuscript or as supplementary data.

Reply 1. The mass-spectrometry data are now shown in the Supplementary Table 2.

Supplementary Table 2 Protein-specific peptide intensities of GBR interaction partners related to Figure 1c.

Query 2. Page 5 line 4: The authors should include some discussion of why there is no functional change in ARAP1 & PIANP KO mice in terms of GBR-mediated presynaptic inhibition. I assume the authors are analyzing ARAP1 & PIANP KO mice via immunofluorescence/immunohistochemistry to see what disruption of these complexes does to GBR localization (likely another paper in progress).

Reply 2. We have analyzed axonal GBR levels in *AJAP-1^{-/-}* and *PIANP^{-/-}* mice (new Supplementary Fig. 5) and added the following information to the manuscript:

P6: *AJAP-1^{-/-}* and *PIANP^{-/-}* mice exhibit normal levels of endogenous GB1 protein in the axons of cultured hippocampal neurons (**Supplementary Fig. 5**), explaining why presynaptic GBR-mediated inhibition in these mice is normal (**Fig. 3a**).

Supplementary Figure 5 Normal axonal GBR expression in *AJAP-1^{-/-}* and *PIANP^{-/-}* mice. **(a)** Top: Immunofluorescence of endogenous GB1 protein in axons of hippocampal *AJAP-1^{-/-}* and WT littermate neurons. Neurons expressing GFP were fixed at DIV10, permeabilized, and immunostained for endogenous GB1 protein (green) and the presynaptic marker piccolo (magenta). GFP served as a volume marker. Merged images show GB1 and piccolo co-localization. Bottom: Intensity grey value profile graphs of GB1 (green) and piccolo (magenta). Normalized GB1 fluorescence refers to the GB1 immunofluorescence intensity normalized to the GFP fluorescence intensity. $P > 0.05$, unpaired t-test. Scale bar 5 μm . Bar graphs show that the GB1 fluorescence and the GB1/piccolo co-localization are not significantly different between genotypes. The number of neurons analyzed is indicated. **(b)** Immunofluorescence of endogenous GB1 protein in axons of hippocampal *PIANP^{-/-}* and WT littermate neurons. Analysis as in **(a)**.

Query 3. Page 7/ Fig 5 (APP/GB1a complexes co-localize with vesicular trafficking proteins in axons). The authors should measure the vesicle speed to see if it is consistent with kinesin-1 (forward & backward) rates: KIF1A particles move anterogradely in hippocampal neurons with an average velocity of 1.0 $\mu\text{m/s}$, and retrogradely at approx 0.72 $\mu\text{m/s}$.

Reply 3. Axonal transport of GBRs and APP is kinesin-1-dependent (Valdes, V. *et al. PLoS One* **7**, e44168, 2012). KIF1A is a kinesin-3¹⁸ (Hirokawa, N., *et al., Nat Rev Mol Cell Biol* **10**, 682-696, 2009) and has not been implicated in axonal trafficking of GBRs or APP. We have measured the anterograde and retrograde velocity of APPmCherry, GB1a-GFP and APPmCherry/GB1a-GFP vesicles. The measured anterograde and retrograde velocities of APPmCherry and GB1a in axons are comparable to those reported in dendrites. We have included the velocity data into the manuscript.

P8: The mean anterograde ($2.2 \pm 0.4 \mu\text{m/s}$, $n = 29$ vesicles) and retrograde ($2.0 \pm 0.2 \mu\text{m/s}$, $n = 29$) trafficking velocities of APPmCherry in axons were similar as reported for dendrites². APPmCherry/GB1a-GFP (antero: $1.6 \pm 0.2 \mu\text{m/sec}$, $n = 16$; retro: $1.3 \pm 0.2 \mu\text{m/sec}$, $n = 26$) and APP-VN/GB1a-VC vesicles (antero: 1.6 ± 0.1 , $n = 19$; retro: $1.5 \pm 0.1 \mu\text{m/sec}$, $n = 8$) had

lower mean transport velocities than APPmCherry alone. GB1a-GFP vesicles (anterograde: 1.0 ± 0.1 , $n = 15$; retrograde: 1.2 ± 0.1 $\mu\text{m}/\text{sec}$, $n = 25$) had even lower transport velocities, similarly as observed before³.

P11: Differences in the transport velocities of GB1a-GFP and APPmCherry/GB1a-GFP vesicles further suggest the existence of an APP-independent GB1a transport.

Query 4. Show some full size images of neurons for CSTN1 & 3, JIP-1b, JIP3 immunofluorescence (also with the KLC1 & VENUS from the APP/GB1) as supplementary data.

Reply 4. We now provide full size images of neurons stained for CSTN1/3 and JIP1/3, Venus and KLC1 in the new Supplementary Fig. 8.

Supplementary Fig. 9 The APP-VN/GB1a-VC BiFC complex (green) partly colocalizes with CSTN and JIP in neurons. Co-localization (white, arrowheads) of the APP-VN/GB1a-VC BiFC complex (green) with FLAG-CSTN-1, FLAG-CSTN-3, FLAG-JIP-1b and FLAG-JIP-3 (cyan) and the endogenous kinesin light-chain 1 (KLC1) (blue) in transfected neurons. Note that KLC1 is expressed in axons and dendrites. Scale bar 10 μm . Higher magnification images of axons are shown in **Fig 5c**.

Query 5. Page 8, 2nd paragraph (Interaction of APP with GB1a inhibits A β generation): why is there no data presented on quantification of sAPP cleavage product formation, only APP- $\beta\text{CTF}/\text{APP-FL}$ ratio? This should be provided.

Reply 5. We have quantified the sAPP/APP-FL ratio and included the data into Fig. 6b.

P8: Densitometric analysis revealed a reduction in the APP- $\beta\text{CTF}/\text{APP-FL}$ and sAPP $\beta/\text{APP-FL}$ ratio by 60% ($P < 0.0001$ vs exogenous APP together with BACE1) and 57% ($P < 0.01$), respectively, in the presence of GB1a/2 receptors (**Fig. 6b**).

Fig. 6 (b) Likewise, a significant reduction in the sAPP/APP-FL ratio is observed in the presence of GB1a versus GB1b (APP: $13.2 \pm 0.4\%$; APP+BACE1: $100.0 \pm 4.1\%$; APP+BACE1+GB1a/2: $43.3 \pm 5.3\%$; APP+BACE1+GB1b/2: $95.8 \pm 5.4\%$; $***P < 0.001$, one-way ANOVA, $n = 3$ independent experiments).

Query 6. A critical additional experiment here is to show overexpression of GB1a in neurons can rescue the increased A β 40 secretion measured in hippocampal neurons (addition to Fig 6d)

We infected cultured hippocampal neurons of *GB1a*^{-/-} mice (DIV3) with lentiviral particles expressing GFP-GB1a or control GFP. We determined A β 40 expression in the conditioned medium at DIV10 using an ELISA. The results show a significant ($P < 0.05$) 15% decrease in A β 40 levels in neurons in the presence of exogenous GFP-GB1a compared to control GFP. We also infected cultured WT hippocampal neurons with lentiviral particles expressing GFP-GB1a or GFP-GB1b. We similarly saw a significant decrease in A β 40 levels in the presence of exogenous GFP-GB1a but not GFP-GB1b. We have included the data into Fig. 6f.

P9: Infection of cultured hippocampal neurons of *GB1a*^{-/-} or WT mice with lentiviral particles expressing GFP-GB1a significantly reduced A β 40 secretion ($P < 0.05$ vs GFP or GFP-GB1b, Fig. 6f).

A β 40 quantification. Transfected HEK293 cells were incubated with serum free DMEM/F-12 medium (Gibco 11320-074) for 24 hours. After determining the total amount of protein in the supernatant the Wako II A β 40 ELISA kit was used for A β 40 quantification. For A β 40 quantification in neurons, 1×10^5 neurons from *GB1a*^{-/-}, *GB1b*^{-/-} or WT littermate mice were incubated for 10 days in conditioned medium. In some experiments purified lentiviral particles (GeneCopoeia: 217LPP-Rn10234-Lv122 GFP-GB1a, 217LPP-Rn10298-Lv122 GFP-GB1b, 217LPP-EGFP-Lv242 GFP) were added at a concentration of 1 transforming unit per neuron after 3 days for 24 hours in 200 μ l conditioned medium, before adding 800 μ l pre-warmed conditioned medium. After 10 days, the conditioned medium was cleared at 1,000 x g for 15 min at 4°C and processed for A β 40 quantification.

Fig. 6 (f) Lentiviral expression of GB1a but not GB1b decreases A β 40 secretion in neuronal cultures of *GB1a*^{-/-} and WT mice. Bar graphs of the amount of A β 40 secreted within 10 days into the culture medium of hippocampal neurons infected with purified lentiviral particles expressing GFP, GFP-GB1a or GFP-GB1b. *GB1a*^{-/-} cultures: GFP 100.0 \pm 10.2%, GFP-GB1a 83.4 \pm 8.9%, * $P < 0.05$, $n = 4$ independent neuronal cultures, $P > 0.05$, unpaired Student's t-test. WT cultures normalized to uninfected (100%): GFP-GB1a 75.2 \pm 1.7%, * $P < 0.05$; GFP-GB1b, 108.9 \pm 16.8%, $P > 0.05$, one-way ANOVA; $n = 3$ independent neuronal cultures.

Query 7. It would be very informative also to know if sAPP β and α are increased in the GB1a^{-/-} neurons, as sAPP β and α have neurotrophic properties and sAPP α knock-in can rescue APP^{-/-} deficits in spatial memory and LTP (Ring et al. 2007).

Reply 7. Commercially available antibodies commonly used to detect sAPPs, including mouse anti-APP A4 22C11 (mab348, Millipore), mouse anti-sAPP α clone 6E10 (SIG-39320, BioLegend), mouse anti-sAPP α clone 2B3 (11088, IBL), mouse anti-sAPP β (Poly 8431, Biolegends) were unable to detect sAPP α and sAPP β in the supernatant of cultured hippocampal neurons of GB1a^{-/-} and WT littermate mice, using Western blots as read-out. In the same cultures, we could detect A β 40 using the Wako II A β 40 ELISA kit. Since we had to make cultures from single embryos of heterozygous GB1a^{+/-} breedings, we were unable to scale up culture conditions to possibly detect sAPPs in the cell medium.

Query 8. Figure 7: The α -bungarotoxin (BTX) binding experiments to measure endocytosis are regularly performed in neurons, and this should also be done here. GB1A receptors have a much longer surface stability (slower internalization) compared to GB1B in hippocampal neurons as shown by the lab of Trevor Smart (Ref 11, Hannan et al'12). This prior finding should also be mentioned in this publication, as it is highly relevant to the data in Fig 7. In 7E, is the endocytosis rate of GB1a with APP significantly different than GB1b or not – no statistics are provided or significance indicated in the figure or legend.

Reply 8.

We have repeatedly tried to do the α -bungarotoxin (BTX) binding experiments to measure endocytosis in neurons. For unclear reasons we did not obtain BTX staining at the cell surface of BBS-APPmCherry transfected neurons, even though we could obtain BTX staining in permeabilized cells and also observed mCherry fluorescence intracellularly (see Figure below). However, we have carried out experiments showing that expression of GB1a-GFP in cultured hippocampal neurons decreases localization of APP-mCherry in early endosomes (Supplementary Fig. 12). This experiment circumvents the use of the problematic BBS construct and similarly shows that GB1a prevents internalization of APP.

We have included a statistical analysis to the internalization experiments in Fig. 7e that shows that the internalization of APP in the presence of GB1/2 receptors is significantly slower than in the presence of GB1b/2 receptors or in the absence of GBRs. We now also refer to previous experiments by the lab of Trevor Smart.

Figure showing that BBS-APP-mCherry cannot be labelled with BgTx-488 at cell surface of neurons. Top: Live imaging of BBS-APP-mCherry in primary hippocampal neurons after 10 min surface labelling with BgTx-488. Neurons expressing BBS-APP-mCherry (magenta) were incubated with 3 $\mu\text{g/ml}$ BgTx-488 (green) for 10 min in Krebs buffer at room temperature at DIV8, washed and imaged. The merged image shows overlay of surface BgTx-488 labelling and overexpressed BBS-APP-mCherry. Bottom: Immunofluorescence of BBS-APP-mCherry in primary hippocampal neurons labelled with BgTx-488. Neurons expressing BBS-APP-mCherry (magenta) were fixed at DIV8, permeabilized and stained with BgTx-488 (green). Merged image shows extensive overlap of BgTx-488 with BBS-APP-mCherry in permeabilized cells. Scale bars 10 μm . Confocal images were processed by Z-projection (Max intensity).

We have made the following revisions to the manuscript.

P10: GB1a/2 receptors exhibit slower internalization and longer surface stability than GB1b/2 receptors in neurons¹⁹. It is therefore possible that APP reciprocally stabilizes GB1a/2 receptors at the cell surface, although we did not directly test this.

We addressed whether the presence of GB1a prevents sorting of APP into endosomes in cultured hippocampal neurons (DIV14) transfected with GB1a-GFP, APPmCherry or both. To identify recycling endosomes we incubated neurons with Alexa-AF647 conjugated transferrin. Co-expression of GB1a-GFP indeed significantly decreased the presence of APP-

mCherry in transferrin-AF657 positive endosomes (**Supplementary Fig. 12**). We further analyzed whether neuronal activity influences APP-mCherry and GB1a-GFP sorting into endosomes. To induce neuronal activity we used an established glycine/bicuculline stimulation protocol². This protocol did not significantly alter endosomal localization of APP-mCherry in the presence and absence of GB1a-GFP (**Supplementary Fig. 12**).

P16: Transferrin-AF647 (Invitrogen) and bicuculline/glycine treatment² was performed at DIV14. 30 min before transferrin treatment, neurons were incubated with fresh Neurobasal medium. Incubation of Transferrin-AF647 at a final concentration of 50 µg/µl was for 1 hour, after which control cultures were kept in pure Neurobasal medium. For activity induction neurons were treated with 20 µM bicuculline/200 µM glycine for 5 min in Neurobasal medium. The medium was then replaced with fresh medium supplemented with 20 µM bicuculline for 15 minutes. Cultures were washed with 1×PBS, fixed for 10 minutes at room temperature in 4% PFA/4% sucrose and mounted on microscope slides with Dako Fluorescence Mounting Medium (Agilent). Samples were imaged on Zeiss LSM880 confocal microscope equipped with Plan-Apochromat 63×/1.4 Oil objective. Collected Z-stacks were quantified using Fiji and Adobe Photoshop CC 2018.

Fig. 7 (e) Decrease of BTX-488 surface fluorescence in (c) over time. $n = 14$ cells per group, 3 independent transfections per group. *** $P < 0.001$, one-way ANOVA

Supplementary Figure 12 Expression of GB1a-GFP in cultured hippocampal neurons decreases localization of APP-mCherry in early endosomes. Transfected hippocampal neurons were incubated with transferrin-AF647 at DIV14 to determine co-localization of APP-mCherry with transferrin-AF647 positive endosomes. Neuronal activity was elicited in parallel cultures using 20 µM bicuculline and 200 µM glycine. GB1a-GFP significantly decreased the presence of APP-mCherry in transferrin-AF647 positive endosomes. Neuronal activity did not significantly alter endosomal localization of APP-mCherry, both in the presence and absence of GB1a-GFP. Binary images indicate the fraction of APP-mCherry (magenta) present in early endosomes (green) as white regions. Scale bars 10 µm. Bar graphs indicate the ratio of APP-mCherry to total APP-mCherry in early endosomes. Statistical analysis was performed with one-way ANOVA and Tukey's multiple comparison test. $n = 4 - 9$ neurons.

Query 9. Discussion. Although the data in this manuscript support the possibility that GBR downregulation in AD may contribute to excitotoxicity and patient memory deficits, evidence of increased GABA release by astrocytes and enhanced GABAB activity in AD has been also

proposed (and GABAB antagonists as potential AD therapeutics). These contrasting viewpoints should be mentioned.

Reply 9.

We now include a discussion of the observed excess GABA release by reactive astrocytes in AD and the use of GBR antagonists as therapeutics.

P12: However, several reports describe a downregulation of GBRs in AD⁵⁻⁷. GBR downregulation is likely a consequence of the disease, for example caused by increased GBR activity due to excess GABA release by reactive astrocytes^{8,9}. Likewise, dysregulated axonal transport, an early pathological feature in AD associated with increased A β production^{10,11}, will reduce the number of GB1a/2 receptors on glutamatergic terminals and promote NMDA receptor-dependent GBR degradation^{12,13}. GBR downregulation in AD^{6,7} may not only increase A β production but also contribute to excitotoxicity and the high incidence in seizures and memory deficits in patients¹⁴, which is supported by the pathology of *GB1a*^{-/-} and *APP*^{-/-} mice. Increased GABA release by reactive astrocytes in AD^{8,9} may help to counteract excess glutamate release and therefore play opposing roles in excitotoxic processes.

P12: GBR antagonists provide another means to stabilize GBRs at the cells surface by preventing GBR degradation^{9,16}. GBR antagonists are already undergoing evaluation as possible AD therapeutics because they promote excitatory neurotransmitter release and enhance cognition¹⁶.

Query 10.

The statement:” We propose that Memantine® stabilizes APP at the cell surface by preventing NMDA receptor-induced GBR internalization ” – is too strong of a statement, and implies experiments to test this have been performed in the manuscript. Statement could be “ thus it is possible that Memantine® stabilizes APP at the cell surface ...”

Reply 10.

This statement has been altered as suggested.

P13: Thus it is possible that Memantine® stabilizes APP at the cell surface by preventing NMDA receptor-induced GBR internalization^{12,13}.

Minor comments:

Fig 3. No information given on numbers of cultures used, only “n=10 neurons”

P6: 6 independent transfections per group.

Fig 5. No information given on numbers of cultures used, only “n=11 neurons”

P7: 4 independent transfections per group.

Fig 7. no scale bar provided in 7C or D

Scale bars have been added

Table 1: add genotypes across top of all pages

The genotypes are now indicated at the top of all pages.

APP^{-/-} mice also show reductions in KCC2, a neuron-specific K⁺-Cl⁻ cotransporter and a corresponding depolarizing shift in the GABA reversal potential (EGABA)

We comment these findings in the context of the phenotypes of APP^{-/-} mice.

P11: Some phenotypes of APP^{-/-} mice may also relate to a down-regulation of the K⁺-Cl⁻ transporter KCC2 and a resulting decrease in GABA_A receptor inhibition²⁰. Interestingly, GBRs and KCC2 are reported to associate with one another and GBR activity reduces KCC2 levels at the cell surface²¹. Loss of GBRs may therefore counteract downregulation of KCC2 in APP^{-/-} mice.

Spelling

P. 8 and Fig 5 legend: anterogradly or retrogradly, should be anterogradely and retrogradely

Corrected

Fig 6 legend: form, should be from

Corrected

Fig 7 legend: # of independent transfections not given, only n=14 cell,

Fig. 7e 3 independent transfections per group

P. 10 lack of AJAP-1^{-/-} and PIANP^{-/-} mice: remove word “lack”

Corrected

REVIEWERS' COMMENTS:

Reviewer #1 (Remarks to the Author):

Authors have added new data on live axonal trafficking of GBR, colocalization with organelle markers, and GBR/APP at presynapse. Though concerns remain on the APP mobility (as shown in the kymos in the suppl figure 10), I believe, this work as a whole should promote further research on neurotransmitter receptor trafficking in neurons. Role of amyloid beta and betaCTF of APP could also be vital for receptor trafficking. Overall this reviewer is happy to recommend the revised manuscript for publication in Nature Communication.

Reviewer #2 (Remarks to the Author):

The authors have addressed all of my critiques appropriately. In combination with their responses to the other reviewers, this manuscript is much clearer and stronger than the original submission. I have no further critiques or concerns.

Reviewer #3 (Remarks to the Author):

Collectively, the authors have satisfactorily addressed most of my previous concerns, and the manuscript has substantially improved. In general, the authors were rigorous and thorough in their efforts to answer all of the reviewers concerns. In summary, this manuscript is highly likely to influence the field, and has major implications for potential treatment development in AD. However, before accepting this for publication, as per my original request and following the Nature policy for proteomic data(see below weblink and quote) and standard expectation in the field, mass spectrometry proteomics raw files and data analysis files should have been deposited to the ProteomeXchange Consortium (<http://www.proteomexchange.org/>) via the PRIDE partner repository and cited appropriately in the paper. I would point out that sufficient data should be provided in order for it to be possible to follow/reproduce the analysis from proteomic raw data to final product. <https://www.nature.com/authors/policies/availability.html> "Mandates for specific datasets For the following types of data set, submission to a community-endorsed, public repository is mandatory. Accession numbers must be provided in the paper."

Reviewer #3

Query. However, before accepting this for publication, as per my original request and following the Nature policy for proteomic data (see below weblink and quote) and standard expectation in the field, mass spectrometry proteomics raw files and data analysis files should have been deposited to the ProteomeXchange Consortium (<http://www.proteomexchange.org/>) via the PRIDE partner repository and cited appropriately in the paper.

Reply. The mass spectrometry proteomics data have been deposited to the ProteomeXchange Consortium via the PRIDE partner repository with the dataset identifier PXD012487 (Project DOI 10.6019/PXD012487). We have added the following information to the “Data Availability” section in the manuscript.

The mass spectrometry proteomics data have been deposited to the ProteomeXchange Consortium via the PRIDE partner repository with the dataset identifier PXD012487 [<https://www.ebi.ac.uk/pride/archive/projects/PXD012487>]